# Spontaneous whole-genome duplication restores fertility in interspecific hybrids

Guillaume Charron [1,2,3,5], Souhir Marsit[1,2,3,4,5], Mathieu Hénault[1,2,4], Hélène Martin[1,2,3,4] &
Christian R. Landry [1,2,3,4]

Interspecies hybrids often show some advantages over parents but also frequently suffer from reduced fertility, which can sometimes be overcome through sexual reproduction that sorts out genetic incompatibilities. Sex is however inefficient due to the low viability or fertility of hybrid offspring and thus limits their evolutionary potential. Mitotic cell division could be an alternative to fertility recovery in species such as fungi that can also propagate asexually. Here, to test this, we evolve in parallel and under relaxed selection more than 600 diploid yeast inter-specific hybrids that span from 100,000 to 15 M years of divergence. We find that hybrids can recover fertility spontaneously and rapidly through whole-genome duplication. These events occur in both hybrids between young and well-established species. Our results show that the instability of ploidy in hybrid is an accessible path to spontaneous fertility recovery.

[1] Institut de Biologie Intégrative et des Systèmes, 1030 avenue de la Médecine, Université Laval, Québec (Qc) G1V 0A6, Canada. [2] Regroupement Québécois de Recherche sur la Fonction, l'Ingénierie et les Applications des Protéines, (PROTEO), 1045 Avenue de la Médecine, Université Laval, Québec (Qc) G1V 0A6, Canada. [3] Département de biologie, 1045 Avenue de la Médecine, Université Laval, Québec (Qc) G1V 0A6, Canada. [4] Département de biochimie, microbiologie et bio-informatique, 1045 Avenue de la Médecine, Université Laval, Québec (Qc) G1V 0A6, Canada. [5] These authors contributed equally: Guillaume Charron, Souhir Marsit. Correspondence and requests for materials should be addressed to C.R.L. (email: Christian.landry@bio.ulaval.ca)

nterspecific hybridization is common in animals, plants, and microorganisms[1,2] and is a potentially frequent source of genetic diversity over short time scales[3–5]. However, hybrid lineages often suffer from poor fertility that reflects reproductive isolation between parental lineages. The poor fertility of hybrids can prevent their maintenance as independent populations, thus hindering their long term and thus speciation potential. Different molecular mechanisms underlie hybrid infertility, including genetic incompatibilities (nuclear and cytonuclear)[6] and changes in genome architecture (ploidy number or chromosome rearrangements)[7]. If the hybrids are to establish as an independent population or species, they need to recover from this low initial fertility. In obligatory sexual species, fertility restoration can be achieved by crosses among hybrid individuals or backcrosses with either parental species, allowing the purge of incompatibilities through recombination. Because of this gene-flow, this process most often leads to the formation of introgressed species[8] rather than hybrid species[9]. In this context, the formation of hybrid species may necessitate other means of isolation from both parental species, which may include geographic or ecological isolation while the recovery of fertility through recombination takes place[10]. Some organisms, however, have access to both sexual and asexual reproduction. In these species, if sexual encounters are rare, periods of asexual reproduction might provide hybrids with alternative mechanisms for fertility recovery, which could facilitate hybrid speciation.

Using yeast as an experimental model system, we show that hybrids between closely and distantly related species can recover fertility spontaneously by whole-genome duplication and this, without the need for natural selection. Although rare, these events have large effects and bring fertility to the levels seen in parental species. Polyploidy is most common in plants[11] but has also been observed in many animals and fungi[12,13], making this mechanism of fertility recovery accessible to many species.

## Results

**Hybrid survival through serial bottlenecks**. We investigated the evolution of fertility in parallel experimental yeast hybrid populations during mitotic evolution. Using strong population bottlenecking to minimize the efficiency of natural selection allowed for the random accumulation of genetic and chromosomal changes, providing an estimate of the neutral rate of evolution. We examined whether fertility would increase or decrease with time and, if so, whether it would occur through gradual or punctuated changes (Fig. 1a). We considered hybridization over different levels of parental divergence from intra-population to interspecific crosses. These crosses represent up to 15 M years of divergence, which is sufficient to achieve almost complete (99%) postzygotic reproductive isolation in budding yeast[14,15]. We used a collection of North American natural yeast isolates representing three lineages of the wild species *Saccharomyces paradoxus* and two wild isolates of its bona-fide sister species, *S. cerevisiae* (Supplementary Table 1). The *S. paradoxus* lineages (*SpA*, *SpB*, and *SpC*)[16] are incipient species that exhibit up to 4% of genetic divergence (*SpA–SpB*)[17,18] and up to 60% of reduction in hybrid fertility compared to within lineage crosses[19]. These species and populations occur in partially overlapping geographical ranges, even for the most distant pair, *S. paradoxus* and *S. cerevisiae*, making hybridization possible in a natural context. Including this sister species extends nucleotide divergence between parental strains up to 15%. We mated two *SpB* strains to two other *SpB* strains and to two strains of the diverged lineages and species, producing four different types of crosses in duplicates that we classified in terms of divergence: Very Low ($VL_{div}$) = $SpB \times SpB$; Low ($L_{div}$) = $SpB \times SpC$; Moderate ($M_{div}$) = $SpB \times SpA$, and High

($H_{div}$) = $SpB \times S. cerevisiae$ (Supplementary Table 2 and 3). Ninety-six diploid hybrid lines from independent mating events were generated for all but the $VL_{div}$ crosses, for which 48 hybrids were generated, for a total of 672 independent lines (96 lines × 3 types of crosses × 2 pairs of strains + 2 × 48 $VL_{div}$ crosses) (Fig. 1b). We randomly selected and streaked colonies on plates every 3 days for 35 passages with an estimated number of mitotic generations of 22 per passage (Fig. 1c, Supplementary Fig. 1).

Not all lines could be propagated through the entire experiment. After 770 mitotic generations, 77.9% of the lines (524 out of the 672 initial lines) were still propagated using standard conditions. About 50% ($n = 72$) of the extinct lines were lost within the first 250 mitotic generations. This suggests that the loss of these lines is mostly due to genomic instability that arises rapidly after hybridization rather than spontaneous mutations, which would happen at a much slower pace[20]. The $L_{div}$, $H_{div}$, and one of the $M_{div}$ crosses (M1) had a significantly lower proportion of surviving lines (averages of 69.3% for $L_{div}$, 70.3% for $H_{div}$ and value of 79.2% for M1, Fig. 1d, Supplementary Fig. 2) compared to $VL_{div}$ and M2 (average of 95.8% for $VL_{div}$ and value of 91.7% for M2, Fig. 1d, $P < 0.01$, Log-rank test, Fig. 1d, Supplementary Table 4). This suggests that hybrids from divergent parents may suffer from exacerbated genomic instability that lead to the rapid collapse of populations when faced with serial bottlenecks. The cause of line extinction remains to be investigated in detail, but the data suggest that it may be because of the frequent segregation of highly deleterious variants generated by genome instability. Indeed, the within-species control crosses of $VL_{div}$, which are expected to be stable because of their low heterozygosity[21] and thus represent a measure of experimental noise, show little line extinction. In general, the extent of line loss correlates with genetic divergence of the parental strains, with the exception of the $L_{div}$ crosses, which shows elevated line extinction. The $L_{div}$ crosses is also the one that shows the greater level of ploidy instability (see the section 'Ploidy evolves following hybridization'), suggesting that these hybrids are generally less stable. Therefore, these results suggest that yeast hybrids, even when maintained by mitotic division only, segregate very unstable clones at high frequency. This is also supported by the fact that the rate of loss decreases with time, which we would expect if unstable genotypes are eliminated through replication. However, replicating multiple colonies from the last available glycerol stocks of some extinct $L_{div}$ and $H_{div}$ lineages over four passages (80 generations) could not recapitulate the high extinction rate we observed during the experimental evolution. This may indicate that the pre-cultures preceding the glycerol stock constitute a selection step favoring the most genetically stable individuals within the colony, even if they are in minority.

**Hybridization rapidly leads to sterility**. We measured two components of strain fertility, first by testing the ability to sporulate and then by measuring spore viability for sporulating strains. We induced sporulation (meiosis) and counted meiotic progeny survival in 214 randomly selected lines at three timepoints roughly corresponding to the initial ($T_{ini}$, right after hybridization, before first passage), middle ($T_{mid}$, 352 generations), and terminal ($T_{end}$, 770 generations) timepoints. For lines that did not survive until the end (58 out of 214), these timepoints were based on the last available frozen stock (Supplementary Table 5). Initial spore survival values were consistent with previous estimates for these types of crosses (averages of <1% for $H_{div}$, 27.7% for $M_{div}$, 34.2% for $L_{div}$, and 60.3% for $VL_{div}$, Supplementary Fig. 3)[19]. However, there were significant differences between the biological replicates (one way ANOVA F(7,552) = 171.8) in the $VL_{div}$ (averages of 47.2% and 73.4%, $P < 0.01$, Tukey

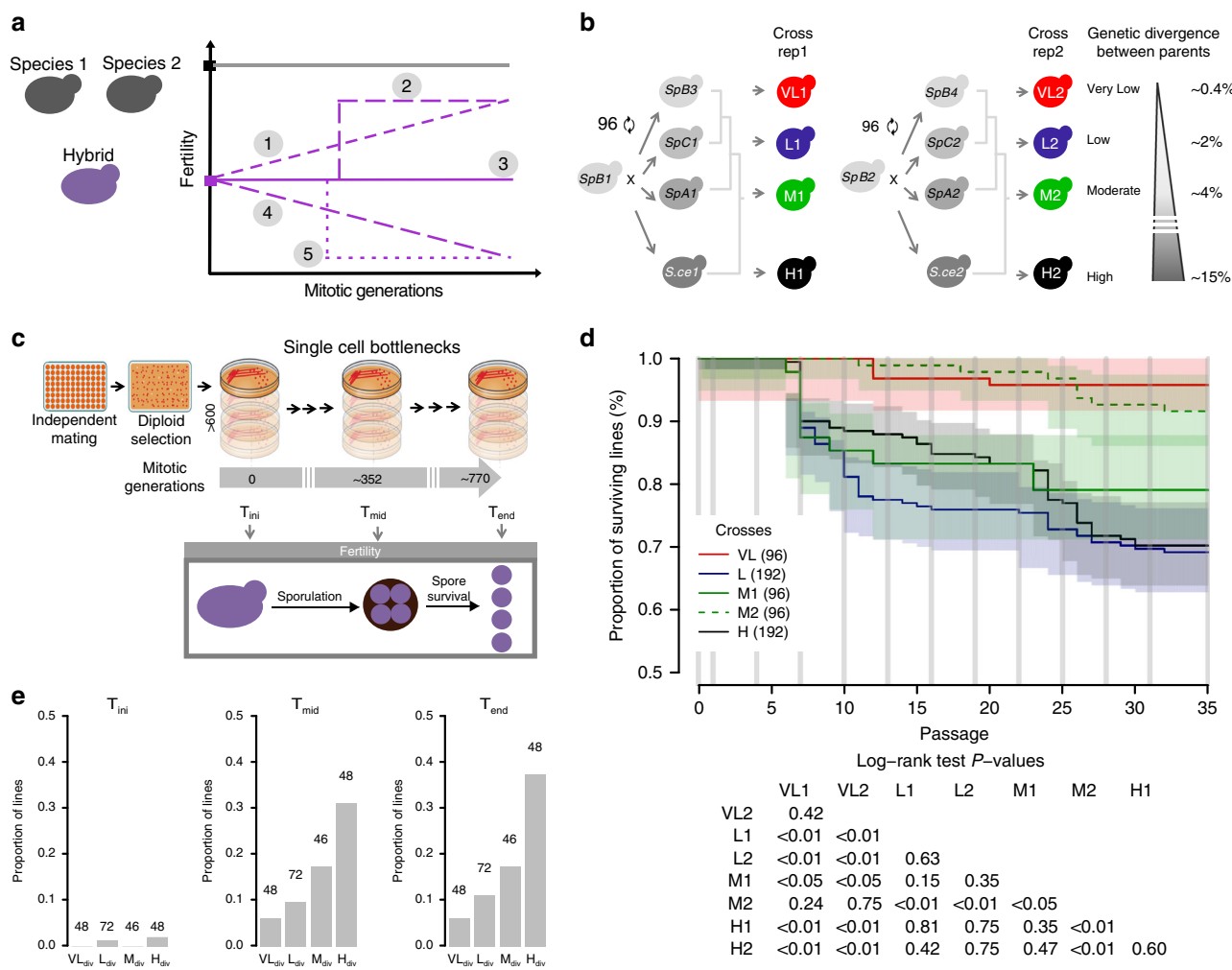

**Fig. 1** Neutral evolution of yeast hybrids and its impact on viability and sporulation ability. **a** Evolution of fertility during mitotic proliferation of hybrids between species: Potential scenarios: (1) Gradual recovery over time, (2) rapid sudden recovery, (3) no significant recovery, (4) decline over time, (5) rapid and sudden decline. **b** Crosses were performed among *S. paradoxus* lineages (VL$_{div}$, L$_{div}$, and M$_{div}$) and with *S. cerevisiae* (H$_{div}$). Each type of cross involves two biological replicates, i.e., involving independent strains, and each individual cross was performed independently to represent independent hybridization events. **c** The 672 hybrids were evolved in conditions of weak selection to examine the neutral spontaneous changes of fertility. Mitotic propagation was performed through repeated bottlenecks of single cells. Fertility was measured by estimating spore viability after meiosis at T$_{ini}$, T$_{mid}$, and T$_{end}$. **d** Survival rates vary among lines. Timing of each glycerol stock indicated by vertical grey lines. Logrank test pairwise comparisons FDR corrected *P*-values are shown. **e** Fraction of lines that lost their sporulation capacity at T$_{ini}$, T$_{mid}$, and T$_{end}$. The number of strains tested per cross type is indicated over the corresponding bars

HSD) and M$_{div}$ crosses (averages of 36.4% and 18.1%, $P < 0.01$, Tukey HSD). These differences are probably due to strain specific genetic variation or even genomic architecture leading to variable levels of postzygotic isolation[19]. Unexpectedly, spore viability could not be assessed for all the lines at T$_{end}$ because 17% ($n = 37$) of the tested lines lost their ability to enter sporulation. We found that the probability of successful sporulation at the end points considered is negatively correlated with parental divergence (Fig. 1e, $r = -.76$, $P < 0.01$, logistic regression). The loss of sporulation ability in yeast is multifactorial[22] but we hypothesized that it could be caused by mitochondrial malfunctions. Indeed, functional aerobic respiration is necessary for sporulation and thus requires the maintenance of functional mitochondrial DNA (mtDNA)[23]. The genotyping of two mitochondrial loci revealed a strong association between the loss of ability to sporulate and the absence of at least one mitochondrial marker (Fisher's exact test, odds ratio > 77, $P = 1.43 \times 10^{-5}$, Supplementary Fig. 4C). This observation suggests that mtDNA instability could contribute to reproductive isolation among closely related yeast populations by

leading to sterility, as shown for more distant species[24], and that this effect increases with genetic distance. There are also rare cases in which lines lost their sporulation ability while both mitochondrial markers were detected (Supplementary Fig. 4), suggesting that the loss of mtDNA integrity is not the only cause of sporulation inability. As the Saccharomyces Genome Database (SGD) reports over 200 genes which, when knocked out in *S. cerevisiae*, lead to an absence of sporulation[25], any loss of function mutation happening in one of those genes could explain this phenotype. Although the *ADE2* gene deletion strategy was used as a color reporter to avoid replicating 'petite' mutants[26,27], it seems that partial or complete mtDNA loss in the hybrid backgrounds did not yield the expected 'petite' phenotype. Despite the fact that lines do not show typical 'petite' phenotypes on glucose, we found a significant association between the absence of sporulation capacity and poor growth on complex media containing glycerol as main carbon source (Supplementary Fig. 5B, logistic regression, $P = 3.97 \times 10^{-10}$), which is consistent with the reduced efficiency of aerobic respiration. Moreover, we also found that

sporulation ability is significantly associated with a change in red pigment saturation for colonies growing on the standard media used for the evolution experiment (Supplementary Fig. 5B, logistic regression, $P = 0.029$). Although the increase of saturation with mtDNA loss was contrary to the phenotype expected from $ADE2$ deletants, this pigmentation change seems to be related with mitochondrial function, as it is well correlated with the ability to grow on glycerol (Supplementary Fig. 5B, Pearson's $r = 0.547$, $P = 1.336 \times 10^{-26}$). These results suggest that the failure of mtDNA maintenance or function plays a major role in causing rapid and irreversible sterility in experimental yeast hybrids through the loss of sporulation ability.

**Fertility evolves through time**. To investigate whether the spore survival component of fertility improved over the experiment, we calculated a fertility recovery score (FRS) as the difference in spore viability between $T_{end}$ and $T_{ini}$ (Fig. 2a, b). As a point of comparison for fertility restoration with sexual reproduction, we performed 12 meiotic generations of intra-tetrad crosses (ITC) in randomly chosen diploid $L_{div}$ lines and calculated FRS, all of which were positive (Fig. 2c, Supplementary Fig. 6). This is in stark contrast with mitotic lines, in which we found no statistically significant bias in FRS values as the distributions were unimodal and centered around 0 (Fig. 2b, Supplementary Fig. 7), showing that spore viability is as likely to increase as it is to decrease. To make sure that low spore viability was not due to the intrinsic inability of strains to produce viable spores, due to dominant mutations for instance, we performed autodiploidization on a random set of 16 haploid spores from the $L_{div}$ and $M_{div}$ crosses and this, at the three timepoints. In most cases, fertility was restored to more than 85% upon selfing (Fig. 2c), showing that infertility mostly derives from the presence of two divergent genomes in the same cell.

Although no general trend towards the recovery of spore viability was observed during the experiment, 23 individual lines presented statistically significant difference in their fraction of viable spores between $T_{ini}$ and $T_{end}$ (Fisher's exact test, FDR corrected, Supplementary Table 6). Eleven of those 23 lines (5 $VL_{div}$, 5 $L_{div}$, and 1 $M_{div}$) showed a decreased fertility and the

remaining 12 (7 $L_{div}$, 4 $M_{div}$, and 1 $H_{div}$) showed improvement. Among the lines that showed improvement, seven (4 $L_{div}$, 2 $M_{div}$, and 1 $H_{div}$) displayed spectacular recovery in spore viability within 352 mitotic generations: their FRS values were close to that of ITC lines (Fig. 2b), and their spore viability was similar to what is typically observed for their non-hybrid diploid parents[17]. One line presented a low FRS value, which is explained by the fact that its fertility returned to its initial value by $T_{end}$ (Fig. 2b). As infertility of *Saccharomyces* hybrids is mainly due to anti-recombination caused by the mismatch repair machinery acting on homeologous chromosome pairs, leading to chromosome missegregation[28,29], there could be at least three main explanations for these sudden increases in fertility. All of these mechanisms rely on the loss of heterozygosity across the genome and re-establishment of correct chromosome pairing during meiosis.

The first potential mechanism would be an endoreduplication event, i.e., spontaneous chromosome doubling following a failed cell division during mitosis[30]. Such an event would lead to the production of identical homologues that would restore correct chromosome segregation. The second mechanism would be damage to a copy of the $MAT$ locus that would convert the diploid hybrid into behaving as a gamete. Two such diploid gametes could then mate, generating a fertile tetraploid hybrid. This path to fertility recovery was recently observed in hybrid species of the *Zygosaccharomyces* genus[31,32], so it is in principle an accessible path to fertility recovery. However, this would need rare events to co-occur in the same colony and to produce two diploid gametes of opposite mating type. The last potential mechanism would be that strains could have sporulated during the experiment and spores of opposite mating types could have mated. This would be the equivalent of our ITC lines where sometimes a single cross between two spores can bring fertility back to high values (Supplementary Fig. 6). This third option is very unlikely because sporulation happens under very specific environmental conditions, principally nitrogen starvation. The frequency of streaking to fresh media during the experiment would prevent such depletion to happen. In addition, this scenario would often lead to spores that are aneuploid, making fertility recovery unlikely even after mating. All these mechanisms

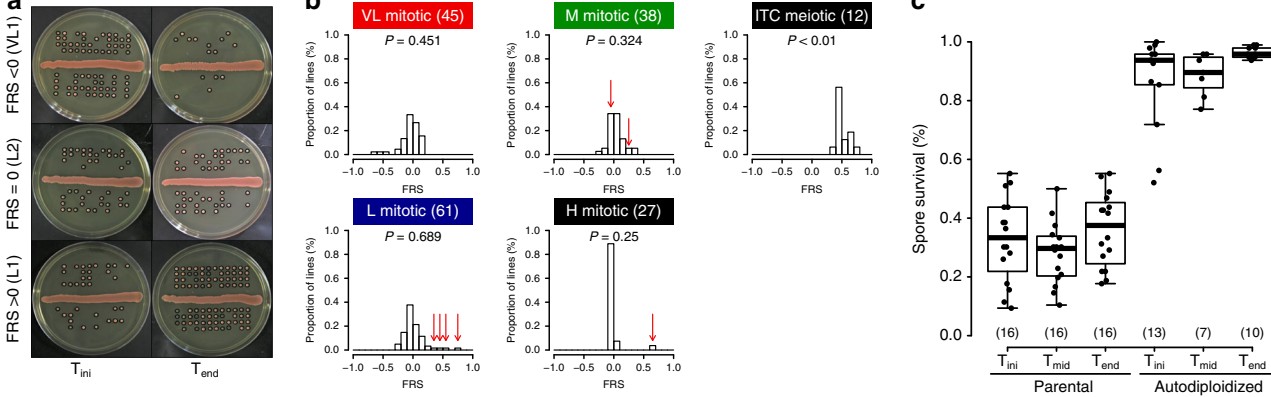

**Fig. 2** Contrast in fertility changes between mitotic and meiotic propagation. **a** Example of spore survival from lines showing a reduction of fertility (top), no change of fertility (middle) and recovery of fertility (bottom). Spores considered as viable are circled in black. Spores from sporulation at the initial time point are shown on the left and from the final time point on the right. **b** Fertility recovery scores (FRS), which measure the change in spore viability, show no clear directionality in mitotically propagated lines ($VL_{div}$, $L_{div}$, $M_{div}$, and $H_{div}$). *P*-values given for exact binomial tests. FRS for intra-tetrad crosses (ITC) indicate large and systematic increase in fertility. Bins containing strains with high fertility by the midpoint of the experiment highlighted with red arrows. **c** Autodiploidization was performed on a random set of 16 haploid viable spores dissected from the $L_{div}$ and $M_{div}$ crosses at $T_{ini}$, $T_{mid}$ and $T_{end}$. Fertility of initial hybrids and evolved lines (parental) increases to above 80% after spore autodiploidization. Numbers in parentheses represent sample sizes. For all boxplots the bold center line corresponds to the median value, the box boundaries correspond to the 25th and the 75th percentile, the whiskers correspond to 1.5 times the inter-quartile range and the dots to individual data points

would generate strains with increased spore viability, but the lines are expected to show a change from diploidy to tetraploidy in the first two scenarios, allowing to differentiate these mechanisms. Mitotic loss of heterozygosity could also be involved but would not be expected to lead to such dramatic recovery of fertility[33]. We tested these potential mechanisms by measuring the total cellular DNA content of the lines to infer ploidy, genome-wide genotyping and whole-genome sequencing of some of the strains.

**Ploidy evolves following hybridization.** We measured ploidy in the 214 randomly selected lines at the three timepoints $T_{ini}$, $T_{mid}$, and $T_{end}$ using DNA staining and flow cytometry. Surprisingly, these analyses revealed that some lines already deviate from diploidy after hybridization. While almost all the hybrids are diploid, both independent $L_{div}$ crosses show frequent triploidy (average at $T_{ini}$ of 54% triploid lines) (Fig. 3a, b, Supplementary Fig. 12). It appears that this triploidization is a major driver of low initial spore viability in the $L_{div}$ hybrids (both crosses), with an average reduction of 45% compared to diploids (20.7% compared to 37.6% considering all timepoints, Supplementary Fig. 8). This triploidy could either be a consequence of aneuploidies that led to an overall DNA content equivalent to triploidy, or as a consequence of whole-genome duplication of one of the parental genomes. We examined the genotype of hybrids using genotyping-by-sequencing (GBS) and found that at $T_{ini}$, all $L_{div}$ triploid hybrids were composed of two copies of the $SpC$ genome

and one copy of the $SpB$ genome (Fig. 3c), suggesting that the change in ploidy predates mating. This would be possible if some $SpC$ haploid cells were in fact diploid. We indeed observed a small fraction of diploid clones in both parental $SpC$ haploid stocks (Supplementary Fig. 9). These clones were identified pseudo-haploids, i.e., diploid but competent for mating (Supplementary Fig. 10). Triploidy appears to be frequent in the $L_{div}$ cross and was observed in the two biological replicate matings performed with independent strains of $SpC$ and $SpB$ (Fig. 1). It however does not seem to occur in all crosses between these two species (Supplementary Figs. 11 and 12) and is variable among replicates (Supplementary Table 7) suggesting some stochastic effects within and between strains. The proportion of triploids among hybrids may depend on the initial proportion of diploid parents (pseudohaploid) in the preculture used for the crosses, which itself is stochastic (Supplementary Fig. 9). The frequency of $SpC$ diploid parents may also depend on whether it appears early or late in the cell culture. The absence of triploids in some other $L_{div}$ crosses indicates that diploidization in parental strains is background-dependent. Phenotypic, genomic, and karyotypic diversity have been observed among strains of the $SpB$ and $SpC$ lineages[17,19]. These results suggest that some $SpC$ haploid strains may be prone to spontaneous genome doubling, the origin of which will need more investigations. For instance, variation that affect key genes involved in cell cycle regulatory pathways could be on the origin of this process.

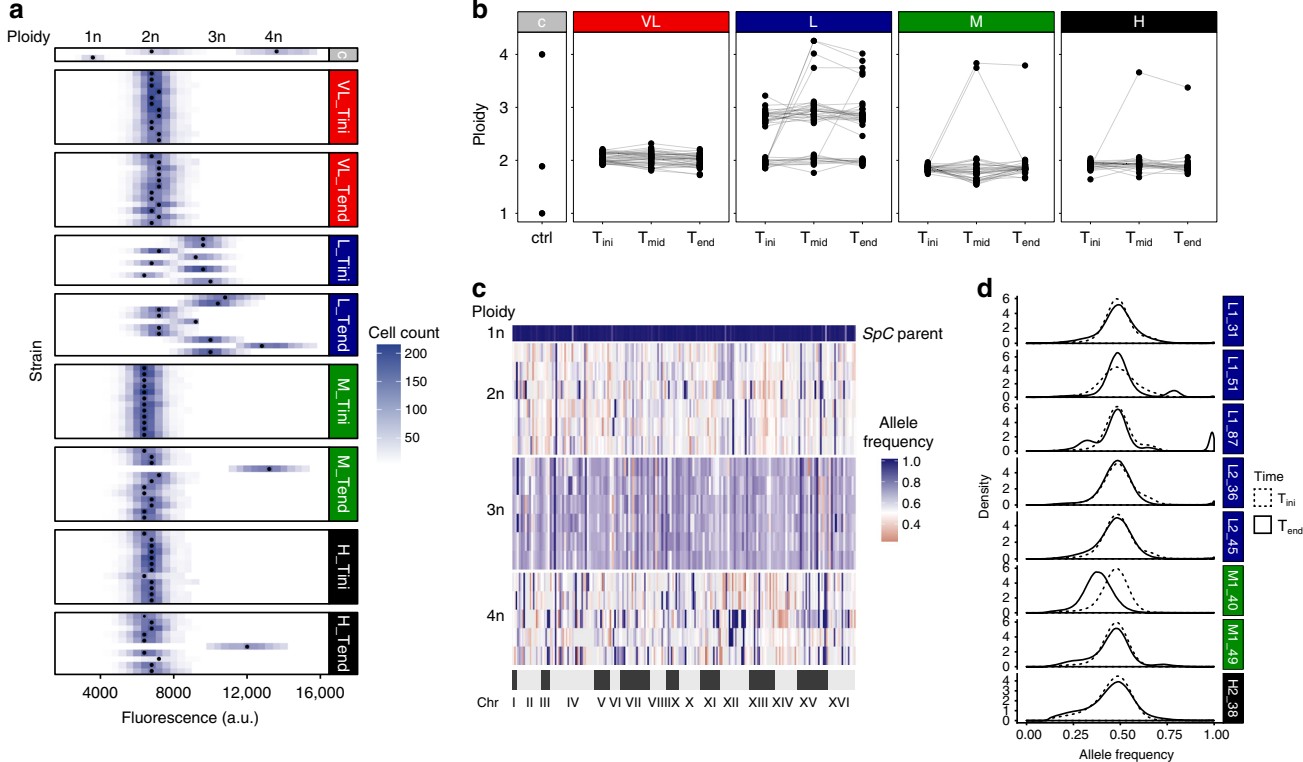

**Fig. 3** Ploidy varies among hybrids and evolves through time. **a** Ploidy of a subset of 12 hybrid lines after hybridization ($T_{ini}$) and after ~770 ($T_{end}$) of mitotic generations. The **c** gray panel corresponds to controls and black circles indicate the highest cell count whose variability suggest that aneuploidy is prevalent in hybrid lines. **b** Ploidy at the three tested timepoints ($T_{ini}$, $T_{mid}$, and $T_{end}$). Connected dots represent independent lines (24). The **c** gray panel corresponds to controls. **c** Frequency of 171 markers corresponding to the $SpC$ parent alleles across the genome of a subset (six diploids, six triploids, and all the five tetraploids) of $L_{div}$ lines show around 50% of $SpC$ alleles in the diploid and tetraploid strains and 66% in triploid strains. **d** Allele frequencies estimated by whole-genome sequencing across the 16 chromosomes. Because diploid hybrids have one copy of each genome, allele frequencies are centered around 50% at $T_{ini}$. This frequency is preserved in most tetraploids, showing the all chromosomes were doubled at $T_{end}$. Loss of heterozygosity is detectable as alternative peaks for instance at 100% (e.g., L1_87) in tetraploids when it involves the loss of one of the two parental alleles. Aneuploidies are also detectable as alternative peaks at 80% (e.g., L1_51) or 30% (e.g. L1_87). M1_40 shows one peak at about 30% suggesting that the sequenced clone of that line is a triploid. The allele frequencies correspond to the hybrid parent 2 alleles ($SpC$ for L lines, $SpA$ for M lines, and $S. cerevisiae$ for the H line)

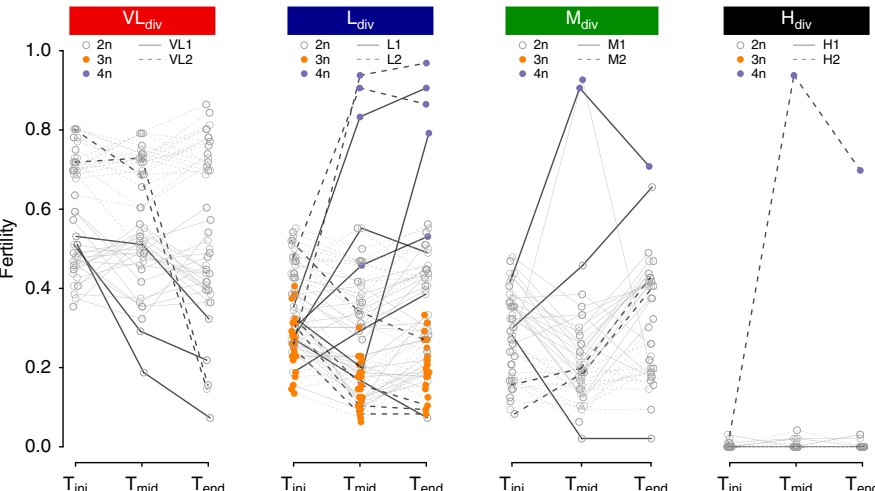

**Fig. 4** Tetraploidization leads to sudden fertility recovery over time. Fertility trajectories at the three timepoints ($T_{ini}$, $T_{mid}$, and $T_{end}$). Each connected set of dots represents an independent line. The colors correspond to ploidy. Dotted and full lines represent the two independent crosses within each genetic divergence class. Bolder lines correspond to the lines with significantly different proportions of viable spores between $T_{ini}$ and $T_{end}$. In the case of VL1 and VL2, the two pairs of strains show different level of fertility, which is frequent in yeast intra-species crosses

Consistent with the hypothesis above based on spore viability recovery by polyploidization, the 7 lines that displayed significant spectacular fertility recovery and one $L_{div}$ line that showed significant fertility improvement, but of lower magnitude, during evolution became tetraploid (Fig. 4). This is also true even for hybrids between *S. paradoxus* and *S. cerevisiae*, which were initially completely sterile (Fig. 4). One of the two tetraploid M1 lines (M1_40) returned to diploidy at $T_{end}$ but this is more likely to be due to segregating ploidy and colony heterogeneity at $T_{mid}$ rather than return to diploidy (for more details see Supplementary Note 2). GBS analysis revealed that tetraploid hybrids have equal copies of parental genomes, since allele frequencies across 171 markers among the 16 chromosomes are around 50% (Fig. 3c, Supplementary Figs. 13 and 14).

As stated above, whole-genome doubling could occur either by endoreduplication, which is a consequence of cytokinesis failure[34] or by means of damage to one copy of the *MAT* locus in the hybrid[30,31] (Supplementary Fig. 15). This damage to the *MAT* locus could cause hybrid cells to behave as a haploid, switch mating type and hence autotetraploidize. In this experiment, mating type switching may not occur using the standard process because the necessary *HO* gene was deleted. The main way by which autotetraploidization could occur by mating in our study is to have two hybrids with damage to the opposite *MAT* loci that are in the same colony and are close enough to mate with each other (Supplementary Fig. 15), which is a very unlikely event. To investigate this, we sequenced the genome of the 8 tetraploid lines (5 $L_{div}$, 2 $M_{div}$, and 1 $H_{div}$) at $T_{ini}$ and $T_{end}$. We indeed found that the frequency of parental alleles across the genomes are roughly 50%, showing that the strains are not aneuploids that would have DNA content equivalent to tetraploidy (Fig. 3d, Supplementary Figs. 16 and 17). One exception is observed for the M1_40 line that show allele frequency corresponding to a triploid state at $T_{end}$ (Fig. 3d, Supplementary Fig. 16) while GBS data show a tetraploid state at $T_{mid}$ and a diploid state at $T_{end}$ (Supplementary Fig. 14). A different colony was isolated each time from these timepoints. This is again consistent with segregating ploidy and colony heterogeneity, which are probably due to extreme genomic instability in this particular line (for more details see supplementary Note 2). Next, we investigated the total or partial chromosome loss or loss of heterozygosity (LOH) of the *MAT* locus region (Supplementary Fig. 15). We identified only two

tetraploids with aneuploidy on chromosome III containing the *MAT* locus (Supplementary Figs. 18, 19, and 20). However, these aneuploidies affect only one copy of the mating type (Supplementary Note 1 and Supplementary Fig. 21). We thus find no evidence of damage to the *MAT* locus of tetraploid hybrids that could have caused mating between diploid hybrids. Thus, these results suggest that endoreduplication is the most likely mechanism of whole-genome doubling. However, we cannot exclude that damage to the *MAT* loci occurring by the loss of chromosome III could be undetectable because this chromosome could have been regained following tetraploid formation. Chromosome III was indeed found to be the most unstable chromosome among *S. cerevisiae* diploids, triploids, and *S. cerevisiae-S. bayanus* hybrids[35]. Furthermore, transient dynamics of aneuploidy were repeatedly observed in yeast laboratory evolution experiments under stressful conditions[36,37]. Such successive aneuploidies could happen in our hybrids lines because aneuploidy is prevalent (Fig. 3c, Supplementary Fig. 18) and tetraploids are notorious for showing genome instability[38].

## Discussion

We measured the rate of recovery from reproductive isolation in yeast hybrids using experimental evolution. We propagated parallel hybrids with serial bottlenecks to measure the neutral rate of recovery. We eliminated the potential confounding effect that natural selection on growth rate could have on genome dynamics and consequently on components of fertility.

One of the most striking changes to fertility we observed is a complete loss of the ability to sporulate within thirty generations, effectively reducing fertility to 0. While decrease in sporulation efficiency was observed in similar experiments performed on homozygous strains[39,40], such complete loss of the ability to sporulate in strains that initially were able to sporulate at high efficiencies was, to our knowledge, not reported before. Our results show that a greater genetic distance between the founding parents led to higher probability of losing the ability to sporulate, suggesting that genome instability or genetic incompatibilities could cause this decrease of fertility. This, added to the results indicating that those strains also lost some or all of their mitochondrial DNA and show growth defects on non fermentable sources, indicate that genetic incompatibilities or instabilities

involving the mitochondrial genome may be responsible for these fertility losses.

We find that reproductive isolation as assessed by spore viability does not have a global directional fate during the mitotic propagation of hybrid lines. The different scenarios presented in Fig. 1a were almost all observed. Overall, spore viability of hybrids evolved like a neutrally evolving quantitative trait, with incremental gains and losses, with no overall particular direction. However, we did observe spectacular punctuated improvements in spore viability for some lineages. From the 23 lines that had statistically significant differences between the initial hybridization and the end of the experiment, fertility decreased for 11 of them. Reductions in spore viability was observed in all crosses except the $H_{div}$ crosses, which were already completely infertile at the beginning of the evolution and could not be reduced further. One explanation for such decrease includes the accumulation of genomic rearrangements, which would lead to incorrect segregation of chromosomes during meiosis to various degrees depending on the extent of the rearrangements[14,19]. The segregation of recessive lethal alleles following a de novo mutation could also be implicated and would lead to strains exhibiting halved fertilities, which were rarely observed, but more complex patterns of fertility decrease are also possible given the potential for genetic interactions in these heterogeneous genetic backgrounds.

Punctual and almost complete restoration of fertility was observed at low frequency in all but the lowest parental divergence crosses. We show that these lines experienced a genome duplication, most likely caused by the doubling of all chromosomes. It was shown before that artificially induced chromosome doubling in infertile hybrids between S. paradoxus and S. cerevisiae hybrids could restore fertility[41]. Our results show that this happens spontaneously, without the need for natural selection and between species that can naturally hybridize. Finally, a more gradual form of statistically significant recovery was also observed in about six lines, which were not subjected to genome duplication. For four of those lines, fertility almost doubled compared to their ancestral lines. The other two have improvement of three and almost five times their initial fertilities. As mitotic recombination often leads to gene conversion events spanning tens of thousands of base pairs[42], accumulation of a large number of mitotic loss of heterozygosity events might allow such recoveries. More in-depth genomic analyses will therefore be needed to understand the basis of these recoveries. This would suggest that the contribution of mitotic recombination to the recovery of fertility in hybrid lineages could be a slow process compared to meiotic recombination and whole-genome duplication.

It is important to note that our results also show that there is some variation for fertility and ploidy (1) within the colonies that were replicated and frozen, (2) among lines within a cross, and (3) between biological replicates of the same divergence category. This variation could be due to the interactions of the parental genetic backgrounds, generating instability in the hybrid lines, instability that starts as early as in the zygotes after the initial mating. Therefore, taking into account the initial genetic background of experimentally evolved hybrids and considering multiple independent hybridization events is crucial for the understanding of their possible evolutionary fates.

Sexual reproduction is thought to be extremely rare in yeast[43] and thus may not always explain successful hybridization and introgression in the wild. The slow rate of improvement of spore viability by mitotic growth alone, without change in ploidy, could therefore contribute to improved fertility because mitotic proliferation is much more frequent than sexual reproduction. This could be even more likely for instance if natural selection could accelerate the loss of heterozygosity, although this remains to be

tested. Our data show that hybrids can recover fertility by becoming allopolyploids in less than 400 cell divisions, which could represent less than 250 days in nature[17]. In principle, fertility recovery could therefore happen before sexual reproduction occurs. The rate of fertility restoration by genome doubling observed in our experiments might be an underestimate of the rate that would occur in yeast hybrids that contain the HO gene. In the event of damage to one of the mating type loci, the wild-type HO would allow the mating type switch and self mating in hybrids, leading to a duplicated genome. How frequently whole-genome duplication occurs in nature remains to be examined but it could contribute to important events. For instance, it was shown recently that the whole-genome duplication[44] in an ancestor of the Saccharomyces genus originated from interspecies hybridization. The success of such an event would have been unlikely if the F1 hybrids were completely sterile and if sexual reproduction would have been needed to recover fertility. Our results suggest that whole-genome duplication could have happened spontaneously and neutrally, thus restoring fertility at the same time. Fertility recovery without sex is likely to apply to multicellular eukaryotes as well because somatic chromosome doubling in diploid tissues or zygotes can lead to the emergence of polyploids, which may display both restored fertility and reproductive isolation with parental species[45]. While polyploidy has been shown to be prevalent in plants[46], it is not common in animals[47,48]. However, the animal lineages containing stable polyploids species often have access to asexual reproductive strategies such as parthenogenesis, which improve tolerance to polyploidy and could enable hybrid fertility restoration without sex[49]. As ploidy changes contribute to restore fertility in partially fertile hybrids, we also find that ploidy instability generates triploids in some of our crosses, which contributes to poor fertility. Ploidy changes are therefore a double-edged sword, causing both reproductive isolation and fertility recovery.

## Methods

**Strain construction.** We used the $ade2$-$\Delta$ marker to help with visual identification of respiration deficient colonies, a strategy used in past mutation accumulation experiments[26,27]. As described in the main text, this marker did not faithfully indicate inefficient respiration of the strain backgrounds that we used. The heterothallic S. paradoxus strains were generated previously[17,19] (Supplementary Table 1). The ADE2 and HO loci of the two wild S. cerevisiae strains were deleted following the method described by Güldener et al.[50]. The ADE2 locus of the S. paradoxus strains were replaced by homologous recombination with resistance cassette following the same procedure as for HO in S. paradoxus[19]. Oligonucleotides with overhangs (Supplementary Table 8) specific to each lineage were used to generate the deletion cassettes from pFA-hphNT1[51] to prevent recombination with the cassettes already present at the HO locus (KANMX and NATMX cassettes).

**Experimental crosses.** Two crosses were made for each of the divergence levels (L1, M1, H1, VL1 and L2, M2, H2, VL2) (Supplementary Table 3). All incubation steps were performed at room temperature (RT). Haploids to be crossed were precultured overnight in 5 mL of YPD (1% yeast extract, 2% tryptone, and 2% D-glucose). Pre-cultures were then diluted at $OD_{600nm}$ of 1.0 in 500 μL aliquots. The aliquots from two strains to be crossed were mixed together and 5 μL were used to inoculate 200 μL of fresh YPD medium in 96 replicates so all strains would derive from independent mating events and would be truly independent hybrids. Cells were given 6 h to mate after which 5 μL of the mating cultures were spotted on a diploid selection medium (YPD, 100 μg mL⁻¹ G418, 10 μg mL⁻¹ Nourseothricin). From each of the 96 spots, one colony was picked as a founding line for the evolution experiment, resulting in 96 independent lines for each of the six interlineage crosses (48 lines for the two intra lineage crosses).

**Evolution experiment.** Each of the independent lines (single colonies) were streaked on one third of a YPD agar plate. To facilitate the detection of potential lines mixing during the experiment, each Petri was streaked with three different crosses (series L1, M1, H1 and L2, M2, H2). Crosses VL1 and VL2 were streaked on two different sets of Petri dishes, with three lineages per Petri. The 192 plates were split into three sets of 64 (lines 1–64 and lineages 65–96). Plates were incubated at room temperature for 3 days after which a new single colony was streaked as a progenitor for the new generation. Each set was rotated between three

manipulators at each replication step. The criteria for the new colony were to be (1) the closest to a predesigned mark on the Petri dish, allowing for unbiased colony selection, (2) a single colony, and (3) big enough to allow for both replication on a new medium and the inoculation of a liquid culture to generate a frozen stock. If the colony closest to the mark did not meet criteria 2 and 3, the second closest colony was then examined, and the process was repeated until all criteria were met. Every three passages, the colonies were both streaked and used to inoculate the wells of a 96-wells plate containing 150 µL of fresh YPD medium. After a 24 h incubation at room temperature, 75 µL of 80% glycerol was added and the plates were placed in a −80 °C freezer for archiving. The lines were maintained on plates for a total 35 passages.

**Estimation of generation time.** To evaluate the generation time on plates and thus estimate the total number of mitotic divisions during the experiment, three lines from each cross were randomly selected. Strains for $T_{ini}$ and $T_{end}$ were thawed (48 total strains tested), streaked on YPD solid medium and let grow for 3 days. Strains were then replicated on fresh YPD solid medium following the same protocol as for the evolution experiment. After 3 days of incubation, the colony closest to the predesigned mark was extracted from the media using a sterile scalpel. The agar block with the colony was put in a sterile 1 mL Eppendorf tube and the colony was resuspended in 500 µL of sterile water. Optical density at 600nm ($OD_{600nm}$) of the resuspensions was estimated using a TECAN Infinite 200 plate reader (TECAN, Männedorf, Switzerland). These resuspensions were diluted in 200 µL of sterile water to obtain $OD_{600nm}$ values of about 0.05 (500 cells µL$^{-1}$). The dilutions were then analyzed with a Guava® easyCyte HT (Millipore Sigma, Burlington, USA) flow cytometer to estimate actual cell numbers. The estimated number of cells µL$^{-1}$ were used to calculate the initial number of cells in the volume used in the dilution and then in the initial 500 µL. The $log_2$ of this number represents the number of cell doubling during the colony growth for one passage of the experiment assuming the colony was formed from a single cell (Supplementary Fig. 5, Supplementary Data 1).

**Sporulation protocol.** Strains were thawed and 2 µL of the stocks were spotted on a fresh YPD medium and incubated for 3 days. A small number of cells was used to inoculate 4 mL of fresh YPD media and incubated for another day. From those pre-cultures, a new 4 mL culture was inoculated at 0.6 $OD_{600nm}$ in fresh YPD and grown for 3 h. Cultures were then centrifuged at $250 \times g$ and the YPD was replaced with 4mL of YEPA medium (1% yeast extract, 2% tryptone, and 2% potassium acetate). Cultures were incubated for 24 h after which they were centrifuged again at $250 \times g$, washed once with sterile deionized water and put into 4 mL of SP medium (0.3% potassium acetate 0.02% D-Raffinose). After 3–5 days of incubation, the strains were dissected as in Charron et al.[19] with a SporePlay™ dissection microscope (Singer Instruments, Somerset, UK) on YPD plates and incubated for 5 days. Pictures of the plates were taken after the incubation and fertility was determined as the number of spores forming a colony visible to the naked eye after 5 days.

**Mitochondrial DNA genotyping.** Two mitochondrial loci were genotyped for presence or absence by PCR. Total DNA extractions were performed using the method described by Looke et al.[52]. The two PCR assays target loci in the *RNL* and *ATP6* mitochondrial genes, respectively, as described in Leducq et al.[53]. Multiplex PCR with both primer pairs (Supplementary Table 8) was performed with the following cycle: 3 min at 94 °C; 40 times the following cycle: 30 s at 94 °C, 30 s at 57.5 °C, and 50 s at 72 °C; and 10 min at 72 °C. A PCR targeting the ITS1-5.8S-ITS2 locus was performed on the same DNA samples as positive controls following the method described in Montrocher et al.[54]. PCR genotyping data and custom scripts used for the analysis can be found in Supplementary Data 2.

**Colony pigmentation analysis.** The stocks of evolved lines at P0, P1, P16, and P35 were printed on OmniTray plates (Thermo Fisher Scientific, Waltham, USA) of YPD solid medium using a BM5-BC-48 colony processing robot (S&P Robotics Inc., Toronto, Canada) and incubated at room temperature for 4 days. Plate images were analyzed with the gitter[55] package in R and positions, which had no growth, had visible contamination, or which were flagged as non-circular or overlapping were filtered out. Downstream analyses were performed using custom scripts in Python 3.7.1. A square of 400 pixels was extracted for each position on the plate and the HSV (hue, saturation, value) were extracted using the package pillow v3.5.0. For each position, the top 10% pixels according to color hue values were retained. Given that hue values allow to discriminate between foreground (colony) and background (media) pixels (Supplementary Fig. 5A), this filtering aimed to minimize the contamination from background pixels. The average of saturation values for each position was used for the analysis. The association between colony saturation and sporulation capacity was tested using a generalized linear model with binomial distribution family and logit link function using the statsmodels package v.0.9.0. Plate images, gitter output files and custom scripts used for the analysis can be found in Supplementary Data 2.

**Colony growth on complex medium with glycerol.** The colonies grown on YPD solid medium for the coloration analysis were replicated on OmniTray plates of

YPG solid medium (1% yeast extract, 2% tryptone, 3% glycerol, and 2% agar) using a BM5-BC-48 colony processing robot and incubated at room temperature for 6 days. Plate images analysis and filtering were performed as described for the colony coloration analysis. A value of one was added to the pixel counts per colony output by gitter prior to conversion in log2. The association between colony size and sporulation ability was tested using a generalized linear model with binomial distribution family and logit link function using the statsmodels package v.0.9.0[56]. Plate images, gitter output files and custom scripts used for the analysis can be found in Supplementary Data 2.

**Fertility assessment in the evolved lines.** Fertility of the evolved lineages ($VL_{div}$, $M_{div}$, and $H_{div}$) was measured on 24 randomly chosen lines per cross (14 that lived through P35 and 10 that were lost before the end of the experiment). To ensure we had the same numbers of diploids to compare with the other lines, more strains were chosen for the two $L_{div}$ crosses, 36 lines were used (24 diploids and 12 triploids, 26 strains that lived until P35 and 10 lost before). For each line, we measured fertility at three different timepoints: (1) immediately after mating ($T_{ini}$), (2) at the halfway point for the given lineage ($T_{mid}$), and (3) at the last passage of the given line ($T_{end}$). This means that $T_{mid}$ and $T_{end}$ do not always refer to 352 and 770 mitotic generations. The information about the generation and fertility data is available in the Supplementary Data 1 file.

**Autodiploidization of spores.** In order to generate fully homozygous strains that should have fully recovered fertility, we performed autodiploidization experiments. After the dissections, some spores from the $L_{div}$ and $M_{div}$ crosses were typed for their mating type locus and their antibiotic resistance markers. When possible, four spores were kept as frozen stocks (two of each mating type and resistance combination). A subset of the spores expressing the G418 resistance was selected to undergo autodiploidization. This was performed by transformation of the spores with the plasmid pHS3 containing *S. cerevisiae HO* gene with its endogenous promoter and the CloNAT resistance cassette (pHS3 was a gift from John McCusker, Addgene plasmid # 81038). Transformants were selected on fresh selection medium (YPD, 200 µg mL$^{-1}$ G418, 100 µg mL$^{-1}$ CloNAT) to be sporulated and dissected.

**Intra-tetrad crosses (ITC).** Intra-tetrad mating was conducted in order to generate hybrids with rapid loss of heterozygosity and to test whether fertility recovery was possible. Strains from the L1 and the L2 crosses were sporulated as described above, but the dissection steps were modified to allow the mating of pairs of spores from the same tetrad (leaving two spores instead of one at the designed dissection spot). The plates were incubated for 5 days. In this manner, pairs of spores of opposite mating types could mate to generate new diploids while pairs of identical mating types would divide mitotically as haploids. To ensure that the yeasts recovered were the result of a mating event, we selected diploids with resistance to both G418 and CloNAT by replica plating the colonies on a fresh selective medium (YPD, 200 µg mL$^{-1}$ G418, 100 µg mL$^{-1}$ CloNAT). From the surviving diploids, one colony was randomly selected to be sporulated again. This process was repeated 12 times (ITC 1 to ITC 12) for 16 lines (2 replicates from lines L1-6, L2-8, L2-36 and L2-63 and 8 other unique line). This number of meiosis was expected to generate extensive LOH as, for a single given heterozygous locus, less than 1% of the population will have maintained heterozygosity[57]. The ITC 0 (initial hybrids), ITC 1, ITC 6, and ITC 12 strains were sporulated and dissected as described above.

**Determination of ploidy.** Measurement of the cell DNA content was performed using flow cytometry with the SYTOX™ green staining assay (Thermofischer, Waltham, USA). Cells were first thawed from glycerol stocks on solid YPD in omnitray plates (room temperature, 3 days) including controls. The parental strain *SpB* (MSH604) was used as control on both its haploid and diploid (wild strain) state. Liquid YPD cultures of 1 ml in 96-deepwell (2ml) plates were inoculated and incubated for 24 h at room temperature. Cells were subsequently prepared as in Gerstein et al.[58], cells were first fixed in 70% of ethanol for at least 1 h at room temperature. RNAs were eliminated from fixed cells using 0.25 mg ml$^{-1}$ RNAse A during an overnight incubation at 37 °C. Cells were subsequently washed twice using sodium citrate (50mM, pH7) and stained with a final SYTOX™ green concentration of 0.6 µM for a minimum of 1 h at room temperature in the dark. The volume of cells was adjusted to be around a cell concentration less than 500 cells µL$^{-1}$. Five thousand cells for each sample were analysed on a Guava® easyCyte 8HT flow cytometer using a sample tray for 96-well microplates. Cells were excited with the blue laser at 488 nm and fluorescence was collected with a green fluorescence detection channel (peak at 512 nm). The distributions of the green fluorescence values were processed to find the two main density peaks, which correspond to the two cell populations, respectively, in G1 and G2 phases. The data were analysed using R version 3.4.1[59].

**Mating type DNA genotyping.** The *MAT* locus was genotyped for the presence of *MAT*a, *MAT*α or both copies by PCR in the stock of the haploid *SpC* parental strains (LL11_004 and LL11_009). Genomic DNA was extracted following standard protocols (QIAGEN DNAeasy, Hilden, Germany) from overnight cultures issued from the five isolated colonies from the *SpC* haploid stocks showing 2n ploidy and as controls three haploid *SpC* strains from the same stocks, two triploid

L$_{div}$ lines, and a diploid SpC control strain (LL11_004). The PCR assay targets a region in the active mating type locus that differentiate the two mating type sequences. PCR with three primers (Supplementary Table 3) was performed with the following cycle: 5 min at 95 °C; 35 times the following cycle: 30 s at 95 °C, 30 s at 55 °C, and 45 s at 72 °C; and 10 min at 72 °C.

**Genotyping by sequencing**. We performed genotyping-by-sequencing (GBS) to investigate the genomic composition of the triploid and tetraploid hybrids. We sampled 77 strains in total: eifht diploids and 16 triploids from the L$_{div}$ crosses at T$_{ini}$; all eight tetraploids and as controls two diploids from each cross L$_{div}$, M$_{div}$, and H$_{div}$ at T$_{ini}$ and T$_{end}$ as well as all parental strains. DNA was extracted from overnight cultures issued from one isolated colony following standard protocols (QIAGEN DNAeasy, Hilden, Germany). As controls, we prepared artificial hybrid genomes by mixing DNA of parental strains with different proportion from each 0.5/0.5, 0.66/0.33, or 0.33/0.66. DNA was quantified using Accuclear® Ultrahigh sensitivity dsDNA Quantification kit (Biotium, Fremont, USA) in a Spark® microplate reader (TECAN, Männedorf, Switzerland). DNA concentration was normalized to 10 ng µl$^{-1}$ and subsequently used for library preparation.

Libraries for Ion Proton GbS were prepared using the procedure described by Masher et al.[60] at the Plateforme d'Analyses Génomiques of the Institut de Biologie Intégrative et des Systèmes (IBIS, Université Laval, Québec, Canada) with the following modifications: ApeKI endonuclease enzyme and ApeKI barcodes were used instead of the PstI/MspI combination and a blue Pippin (SAGE science, Beverly, USA) was used to size libraries (150–300bp) before PCR amplification. Libraries were prepared for sequencing using an Ion Chef™, Hi-Q reagents, and PI™ chip kit V3 (ThermoFisher, Waltham, USA) and the sequencing was performed for 300 flows. A single fastq file was obtained and demultiplexed using Radtags tool from STACKS v1.44 with default options[61], which generated separated fastq files for each sample. Reads were mapped onto the S. paradoxus reference genome (CBS432)[62] for all crosses and onto the S. cerevisiae reference genome (YPS 128)[62] for the H$_{div}$ crosses using Bowtie2 v2.1.0[63]. Read coverage for each position in the genome was also determined using SAMtools v1.8 depth. Single nucleotide polymorphism (SNP) calling on these GBS libraries was performed using the BCFtools v1.4.1 from SAMtools v1.8[64] with default parameters. The data were subsequently analyzed using R version 3.4.1[59]. To measure the allelic frequency, we filtered for SNPs with more than 10× coverage and corresponding to positions in the genome covered in both parental strains to be able to identify the parental origin of the alleles in hybrids.

**Whole-genome sequencing**. We performed whole-genome sequencing to investigate the genomic composition of the eight tetraploid hybrids at T$_{ini}$ and T$_{end}$ (for 16 total) as well as the six corresponding haploid parental strains. Genomic DNA was extracted from overnight cultures issued from one isolated colony following standard protocols (QIAGEN DNAeasy, Hilden, Germany). Libraries were prepared with the Illumina Nextera kit (Illumina, San Diego, USA) following the manufacturer's protocol and modifications from by Baym and colleagues[65]. Pooled libraries were sequenced in paired end, 150bp mode on different lanes of HiSeqX (Illumina, San Diego, USA) at the Genome Quebec Innovation Center (Montreal, Canada). The 22 genomes were sequenced with an average genome-wide coverage of 90×. Raw sequences are accessible at NCBI (bio project ID PRJNA515073).

**Read mapping and variant-calling**. Raw reads were mapped on the reference genome of MSH604[18], one of the two used SpB parental strains, S. paradoxus reference genome (CBS432)[62] for all crosses and onto the S. cerevisiae reference genome (YPS 128)[62] for the H$_{div}$ crosses using bwa mem v0.7.17[66] with default settings. Raw reads were also mapped using Bowtie2 v2.1.0[63] with default settings and the local alignment option on a generated S. paradoxus MAT locus reference sequence for copy number variation and allele frequency of MATα and MATα specific sequences. The MAT locus reference was created by extracting the sequences of MATα and MATα with ±1kb flanked regions from S. paradoxus reference genome (NRRL Y-17217)[67]. Sequences were sorted and indexed with Samtools v1.8[64]. Coverage and variant-calling were performed as for the GBS analysis described above.

**Statistical analyses**. Survival curves were produced and analyzed using the R packages survival[68] and survminer[69]. The analysis of the correlation between sporulation and genetic divergence was performed using the glm R function to perform a logistic regression with the formula: Sporulation T$_{end}$ ~ Genetic Divergence. Statistical analyses and figure creation for fertility data were done using custom scripts (Supplementary Data 1) in R version 3.3.2[59]. Figures and statistical analyses for mtDNA loss and sporulation capacity were performed using custom scripts in Python (version 3.6.3).

***ade2-Δ colony coloration phenotype***. Although we used the ade2-Δ marker as a visual aid to track loss of mitochondrial DNA, we still passaged strains that seem to have lost mitochondrial DNA. During the experiment, some colonies from all crosses suddenly turned whitish or light orange. This paler coloration did correlate with the absence of sporulation in the strains. The only thing that changed during the evolution experiment is the yeast extract (EMD millipore, Burlington, USA),

for which the lot of number changed. Further testing suggests that the pink/red coloration of the ade2-Δ mutants is media dependent. On one of the yeast extract lot used, white colonies appear red and show slower growth while on the other, most colonies are white and show normal growth. The slower growth is common to all ade2-Δ strains (Supplementary Fig. 22).

**Reporting summary**. Further information on research design is available in the Nature Research Reporting Summary linked to this article.

## Data availability

Raw sequencing data that support the findings of this study have been deposited in Sequence Read Archive (SRA), NCBI with the BioProject accession code PRJNA515073. Flow cytometry data that support the findings of this study are available in figshare with the identifier Ploidy_data-Charron_Marsit_2019 [https://doi.org/10.6084/m9. figshare.7877831]. The source data underlying Figs. 1d-e, 2b-c, 3a-c, 4 and Supplementary Figs 1, 3, 4, 6, 7, 8, 9, 10, 11, 12, 13, 14, 18, 19, 20, 21a-b and 22 are provided in the Source Data file. All yeast strains are available from C.R.L. under a material transfer agreement.

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

## Acknowledgements

We thank the members of the Landry lab for discussions and J. Hallin, A. K. Dubé, C. Eberlein, A. Fijarczyk, N. Aubin-Horth, J.B. Anderson, L. Kohn, S. Otto, C. Mérot, and A.M. Dion-Coté for useful comments on the manuscript. We thank C. Mérot for help with the library construction protocols and A. Fijarczyk for help with genome data analysis. This work was supported by grants from the NSERC Discovery and Canada Research Chair to C.R.L, FRQNT scholarship to G.C. and M.H., NSERC Alexander Graham-Bell scholarship to G.C. and FRQS post-doctoral fellowship to S.M.

## Author contributions

Conceptualization: C.R.L., G.C., M.H., and S.M.; Data curation: G.C., S.M. and M.H; Funding acquisition: C.R.L.; Experimental work, formal analysis, and interpretation: Cell propagation by G.C., S.M. and M.H., Fertility by G.C., Ploidy and GBS by S.M. Genome sequencing by S.M. and M.H., Mitochondrial DNA work by M.H., R scripts support for ploidy/GBS by H.M. Project supervision: C.R.L; Writing—original draft: G.C. and S.M.; Writing—review and editing: all authors.

## Additional information

**Competing interests:** The authors declare no competing interests.

