## [Peer Review File · Nature Communications]

Reviewers' Comments:

Reviewer #1:

Remarks to the Author:

This study uses *Saccharomyces* yeast to show that fertility recovery in distant hybrid crosses can take place due to whole genome duplication without sexual recombination. The findings presented are novel and interesting for a broad audience interested in speciation and genome evolution.

I have some comments, questions, and suggestions (in no order of importance) for improvements that the authors may consider. First, a more general point. In my opinion, the narrative of the paper focuses too much on the increasing genetic distance between the crosses made for this study. It forces the reader to have expectations about genetic incompatibilities increasing with the increasing genetic divergence in intra- versus interspecific crosses. I think this sets the reader off on the wrong foot and does not benefit the communication of your most interesting finding, which is that whole genome duplication of parental genomes can restore fertility, even in distant hybrid crosses. In addition, sorting crosses into categories of VL, L, M and H distance classes may mislead readers into believing that the distribution of your crosses is continuous along the genetic distance axis, which is not the case. There is a large gap between the intraspecific (4% divergent) and the intraspecific (15% divergent) crosses. Given the tight word limit you are given, it might be better to tell the reader upfront that your parental lines differed in their initial ploidies and that that is in fact what determined the vastly different outcomes in fertility you observed. In the end, genetic distance played only a minor role in determining the chances for fertility discovery in your study, while ploidy overrides all other effects.

Line 22: "which can hinder their potential as new species or populations." Consider rephrasing. It is not clear what you mean with "potential as new species".

Line 19-31: I think the evolutionary logic of this argument here is not quite right. So because some species have access to both sexual and asexual reproduction, their hybrids can restore fertility via genome doubling. The way it is phrased here, it sounds as if the reverse argument would be that obligatory sexual species do not have access to whole-genome doubling (which is of course not the case as you say at the end of the abstract). As a result, this reads a bit as if evolution in non-obligatory sexual species is intentionally working towards hybrid speciation (while really hybrid speciation is more a by-product of genome doubling). Also, I think the comparison to obligatory sexual species in this context does not really work since in obligatory sexual species, backcrossing with parents usually leads to introgression, but not to hybrid speciation. Instead, to make this comparison valid, why not introduce the concept of homoploid hybrid speciation here and say that in obligatory sexual species homoploid hybrid speciation is rare and usually requires other forms of isolation (e.g. ecological or geographic) of the hybrids from the parents, in addition to restoration of fertility in hybrids through recombination. I think comparing your case with homoploid hybrid speciation sets the stage better for you to show that whole genome duplication in non-obligatory sexual species provides another way for hybrid speciation to work.

Line 44-55: I think it would be fair to mention early on here, that the genetic distances of the crosses you made do not cover the 15M years of divergence evenly. Instead, the majority of your crosses cluster around very small to small distances (0-4%), then there is a big gap, and only then come the large interspecific $S_p \times S_c$ distances (15%). This fact becomes masked in your categories of VL, L, M and H, which may mislead readers into believing the distribution of crosses is continuous along the genetic distance axis. This is especially evident in Figure 1B and 1E where the bar plots give the impression of equal intercepts of genetic distances between the four crosses.

Line 52: Consider rephrasing "in terms of DNA sequences".

Line 69: Your results here are not in conflict with references 13, 14 and 15. These studies have all shown that a small fraction of the hybrid offspring of inter-specific yeast crosses have higher fitness

than the parents and therefore may harbor some adaptive potential, but these transgressive hybrids are not representative of the average fitness of the cross.

Line 71-72: "Hybrids could also have access to more deleterious genomic changes". I think this is a strong understatement. Most hybrids from divergent crosses will on average be less fit than the parental pure-bred crosses as hundreds of studies on hybrid break-down or outbreeding depression have shown, in many different (mainly eukaryote) species.

Line 78-79: The loss of the ability to sporulate is a rather common outcome in yeast strains subjected to experimental evolution (and is probably why laboratory strains like S288c are bad sporulators). That doesn't take away from your findings but should probably be mentioned here.

Line 79: I don't find the overall negative relationship between sporulation efficiency and genetic distance of the cross surprising. That is what we would expect from more and more divergent genomes, i.e. increasing sterility with increasing distance. The most interesting and unexpected element here is that the blue cross (low divergence) has much lower survival than the two more divergent crosses (Figure 1D). I think that is what should be emphasized over all other results here.

Figure 1E. The black and white in the stacked bars is not explained in the figure legend. What am I looking at here? This is the first time it is mentioned that two different parental lines were used.

Figure S2: Similarly here, it is not immediately obvious to me what set 1 and set 2 are since this is not explained in the figure legend.

Line 83: Are there alternative explanations for bad sporulation capacity (other than loss of mitochondria) that should be mentioned here?

Figure 2B: It would help readability of the figure if you made clear on the actual figure that graph 1-4 are a result of mitosis, and graph 5 is the result of meiosis.

Line 92: Please explain if these are ITCs from all 4 cross types (VL, L, M and H). Given Figure S6, it looks that all ITCs are derived from SpB x SpC crosses? I also believe that the figure reference should be to Fig. 2B not C.

Line 96: I am not sure I understand this test. You autodiploidized (selfed) these strains to make them fully homozygous. So this tests for the presence of any intra-chromosomal (homologous) incompatibilities between the two divergent genomes present in the hybrid. At the same time this controls for potential effects of antirecombination, preventing crossing over and leading to the missegregation of chromosomes during meiosis, which can result in aneuploidies. However, autodiploidization does not exclude potential negative epistatic effects between non-homologous chromosomes, which may also lead to low spore viability. Perhaps I am misunderstanding something here but this may need some more explanation.

Line 101-105: Please present some statistical evidence and add a figure for this result: "...we observed cases of spectacular fertility recovery in seven mitotic lines (4 Ldiv lines, 2 Mdiv lines and 1 Hdiv line), which displayed FRS close to that of ITC lines." If these are the seven lines shown in the histogram in Figure 2b (the outlier bins) it might be a good idea to highlight this in the histogram.

Line 112: It is not clear (to the non-yeast scientist) why autodiploidization is less likely an explanation than autopolyploidization here.

Figure 3 C: the variation along the y-axis indicates that ploidy was not an exact measure (i.e. ploidy did not come out exactly as 2N, 3N or 4N). You don't mention the possibility that some of these strains may have aneuploid chromosome content at all. Is it possible that you have seen many different aneuploidies come and go in the genomes over the course of experimental evolution (as e.g. seen in Yona et al. PNAS 2012)? It would be good to see a discussion of this somewhere in the paper.

Line 136: There appears to be many more than 6 lines showing more than 70% fertility after evolution in Figure 4. The remaining lines, outside of the 6 that you mentioned, were all in the VL divergence cross. These lines all remained diploid. I think it would be important to mention this because the way it currently reads is that the only possibility of maintaining >70% fertility is to become tetraploid.

Also, in Figure 4 in the VL crosses it appears that there was a difference between the two independent crosses (dotted and full lines). Is this difference significant and can it be explained? Lastly, in Figure 4 it appears that in the M crosses there was a line that was tetraploid at Tmid but became diploid at

Tend. This would be worth mentioning.

Line 148: I don't follow the argument in this sentence: "Mitotic fertility recovery contrasts with our results and that of others showing that sexual reproduction leads to incremental and rapid recovery of fertility". It is unclear what argument is being made here. As I understand it, this study showed that mitotic fertility recovery is possible, so therefore how does it contrast with your results?

Line 160-163: Consider specifying here that the "multicellular eukaryotes" you used to support your statement about fertility restoration without sex are in fact plants. It would be good to see a discussion here on whether you think WGD would also be a functional mechanism for the restoration of fertility in animals as you mention in the abstract. I think it would be difficult to find many supporting references for fertility recovery without sex in animal hybrid crosses.

Line 330: Replace "cross" with "crosses".

Line 345: Should this be Table S2?

Rike Stelkens

Reviewer #2:

Remarks to the Author:

The study by Charron et al. examines neutral fertility recovery among hybrid strains of varying divergence levels passaged through repeated mitotic bottlenecks for ~800 generations. Interestingly, although the initial probability of successful sporulation was negatively correlated with parental divergence, the recovery of fertility (measured through spore viability assays) evolved neutrally – strains were generally equally likely to gain or lose fertility. The exception was ploidy-variant strains: triploid strains were more likely to have low fertility, while tetraploid strains were more likely to have high fertility. I generally found this to be an enjoyable read of an interesting and novel story about the interplay between strain divergence and ploidy on neutral recovery of fertility following hybridization.

Based on the format and brevity of the text, it seems that this manuscript was previously submit to a journal with more strict word limits, and not revised prior to submission here. I appreciate that the format should not need to be changed prior to re-review here (and indeed strongly believe that initial review at all journals should accept manuscripts in any format the authors choose). Many of my concerns however are as a direct result of (my perceived) previously-imposed brevity and my overarching request is that the authors better analyze and discuss the intricacies of the interesting data they have collected.

Major questions

1. A nice component of this work is that the experiments were carried out on two independent crosses for each divergence class, and it is of general interest to know how strain background differences influences the results. Hence, it was surprising to see that the authors did not generally present the data or provide a narrative about the similarities or differences in results from different crosses of the same divergence class. Since there is internal structure of strain relatedness, it seems critical that the data not be collapsed together without first statistically demonstrating that there were no significant differences between crosses of the same divergence class (e.g., in sporulation capacity, survival, ancestral spore viability, etc.) and/or taking into account the relatedness structure in analyses. The data in figure 4 (initial fertility) is plotted (but not discussed) by background and there does seem to be a background effect for VL and M strains but not for L?

2. The survival (extinction) data is interesting, particularly in contrast with the heterosis has been previously reported with large populations. I found the language to be a bit misleading (L69) since

these results aren't really in contrast, and as stated on L71 it really just indicates that the hybrid populations are incredibly diverse, and have access to both more deleterious and more beneficial mutations. In the vein of presenting all the data, it would be good to have a table that indicates which strains went extinct (was it equally likely from both backgrounds?). I'm wondering whether the authors explored intra-population viability/probability in survival- if you streak multiple colonies from the same population that went extinct, how often do you observe extinction vs. survival?

3. Aneuploidy is not discussed but is potentially prevalent. Is everything in Figure C3 euploid? The variation in Figure 3A would suggest to me that aneuploidy is prevalent in VL_Tini and H_Tini strains (as well as potentially all of the L_Tini strains), but if the GBS data shows euploidy then perhaps the mean or median black squares in Figure 3A (what this is should be specified in the figure legend) is misleading.

4. The result that a small fraction of diploid cells in the parental stock led to 50% triploidy in the initial crosses in only some backgrounds seems to suggest a conditional mating advantage for diploids that is background/context dependent. It would be nice to see this fleshed out a bit. Is it just coincidental that both crosses used for the evolution experiments (L1 and L2) exhibit the high prevalence of triploidy when the others do not?

5. A suggested discussion topic/framework is to tie the results back to Figure 1A - you have nicely set up model scenarios for fertility recovery and could easily describe your results in the context of this framework.

Minor concerns

L49 Genetic divergence between SpA x SpB and SpC x SpB is presented in Figure 1B and could be referenced here (until I saw this figure I also wondered about divergence among the different paradoxus lineages). How much divergence is there among different strains of the same lineage?

L49 "up to 60% reduction in fertility in hybrids" -> this is confusing. Reduced fertility in which hybrids and relative to what?

L53-58: I found the section in the text that described the crossing scheme quite confusing and it took me awhile to reconcile Figure 1B and Table S2. It would be helpful to keep the nomenclature consistent throughout. The abbreviations S.pa1 and S.pa2 are unique to this figure panel, I suggest using SpA, SpB and SpC here instead. There are also extra strains listed in Table S2 that were not used in the evolution experiment. Of very minor note, the MSH604 x LL12_028 cross listed in Table S2 should be moved above the UWOPS-91_202 lines.

Figure 3: What is the X axis ("fluorescence") scale?

Figure 4: There seems to be a drop in fertility for Tmid, at least for M1 strains and possibly L1 strains. Is there a biological or methodological explanation for this? If methodological, does this impact other results?

Reviewer #3:

Remarks to the Author:

Charron et al constructed interspecies hybrids between *Saccharomyces* species with various levels of genome sequence divergence. They then passaged the hybrids through ~800 mitotic generations on solid agar media, with random colony choice (bottlenecking) to minimize selection. They analyzed the

fertility (meiotic fitness) of each hybrid at the beginning, middle and end of the experiment. Most of the hybrids had low fertility and there was no significant trend in how fertility evolved (equal numbers of lines decreased and increased their fertility; Fig. 2B). However the major result, emphasized in the manuscript's title, is the discovery that a small proportion of the hybrids regained high fertility by undergoing spontaneous duplication of their whole genome. That is, the initial cross was $1n \times 1n$ (haploids from two parental strains mated), forming a $2n$ hybrid (containing one copy of the genome from each parental species), which then spontaneously became $4n$ and fertile (with 2 copies of the genome from each parental species). The dramatic increase in fertility is attributed to the ability to pair homologous rather than homeologous (sequence-divergent) chromosomes. The proportion of hybrids showing this change was about 3% (7 of 214 hybrids examined).

The manuscript's strengths are (1) that it provides a direct demonstration that meiotic (sexual) fertility can be restored during a period of mitotic (asexual) growth in yeast hybrids, and (2) that it shows that genome doubling is the most potent mechanism of fertility restoration. Moreover, (3) it shows that fertility can be regained very quickly in evolutionary terms (a few hundred mitotic generations). These results are consistent with current opinions about how the ancient whole-genome duplication (WGD) occurred during evolution of an ancestor of budding yeasts. Furthermore, the process that Charron et al observed in yeast should in principle be applicable to any unicellular eukaryotic species that, like yeast, can reproduce both vegetatively or sexually, so the process may be a widespread evolutionary mechanism.

The manuscript's weaknesses are that it provides little new insight into the *molecular mechanism* of fertility restoration. The observation that WGD restores fertility in the hybrids is not particularly surprising because Greig and colleagues (ref. 21) previously showed that artificially causing WGD in an interspecies hybrid restores its fertility. The surprise in Charron et al's work is that WGD occurs spontaneously at a significant frequency. But what is the mechanism of spontaneous WGD?

What this manuscript lacks is insight into the mechanism of spontaneous WGD. Such insight could be obtained by more analysis of the genomes of the fertile interspecies hybrids made by Charron et al. Previous analyses of the genomes of fertile natural interspecies *Zygosaccharomyces* hybrids proposed that their mechanism of fertility restoration was damage to one copy of the MAT locus (PMIDs 28510588 30052970 28842546). This process essentially converts a hybrid zygote into a gamete, enabling it to mate. Did something similar happen in the fertility restoration events that Charron et al observed? My opinion is that the current manuscript is incomplete without an examination of the MAT loci of the fertility-restored strains.

On a related point, if I understand correctly, the strains that were crossed to make the hybrids were all $ho\Delta$, i.e. unable to switch their mating type. While it may have been necessary to design the experiment this way, it means that the hybrids lacked the single most powerful tool they could have used to double their genomes (HO endonuclease). Therefore, the rate of fertility restoration by WGD observed in these experiments may be a substantial underestimate of the rate that would occur in interspecies hybrids that contain an HO gene. At a minimum, this issue needs to be discussed.

Minor comments

L106, This point was recently confirmed by Rogers et al, PMID 30419022.

L109-113. The text here is not very clearly written and I think that readers would struggle to understand the difference between the two explanations, particularly because line 112 says they are "equivalent". Is explanation #1 an endo-reduplication model, i.e. DNA replication without cell division? In explanation #2, are you proposing that the spores have $2n$ ploidy, i.e. that they were produced

without meiosis? Perhaps a picture would help.

Line 114 says that you “tested these hypotheses”, but the test you carried out is a test of whether the (identical) prediction of both of the hypotheses (i.e. 4n ploidy) is verified. The hypotheses were not tested individually. In fact, the MAT locus analysis I suggested above could differentiate between them.

L102 mentions 7 mitotic lines with high fertility (4L, 2M, 1H), but L136 then refers to “6 lines showing more than 70% fertility after evolution”. Looking at Fig 4, it’s not immediately obvious which lines are the 6 in question. It seems that one of the M lines had high fertility and was 4n at the middle timepoint, but at the end timepoint it lost fertility and its ploidy changed (??). If this is correct, the ploidy decrease in this strain needs some more explanation/discussion. It would help if you named the 4n strains.

L148: “Mitotic fertility recovery contrasts with ... results ... showing that sexual reproduction leads to ... recovery of fertility”. Why do you call this a contrast? There is no reason to expect the two processes to be mutually exclusive.

In Fig 3C, why are only four 4n strains are shown, of the 6-7 that were identified? How are we supposed to know which ones are shown?

Reviewers' comments:

Reviewer #1 (Remarks to the Author):

1.1 This study uses *Saccharomyces* yeast to show that fertility recovery in distant hybrid crosses can take place due to whole genome duplication without sexual recombination. The findings presented are novel and interesting for a broad audience interested in speciation and genome evolution.

Answer 1.1: we thank Dr Stelkens for her overall positive evaluation of our work.

1.2 I have some comments, questions, and suggestions (in no order of importance) for improvements that the authors may consider. First, a more general point. In my opinion, the narrative of the paper focuses too much on the increasing genetic distance between the crosses made for this study. It forces the reader to have expectations about genetic incompatibilities increasing with the increasing genetic divergence in intra- versus interspecific crosses. I think this sets the reader off on the wrong foot and does not benefit the communication of your most interesting finding, which is that whole genome duplication of parental genomes can restore fertility, even in distant hybrid crosses.

Answer 1.2: We have now reworded the passages where we talk about this issue so that we do not talk about a gradient of divergence but rather different levels of divergence (see below).

1.3 In addition, sorting crosses into categories of VL, L, M and H distance classes may mislead readers into believing that the distribution of your crosses is continuous along the

genetic distance axis, which is not the case. There is a large gap between the intraspecific (4% divergent) and the intraspecific (15% divergent) crosses.

Answer 1.3: We changed figure 1B accordingly. We now show a discontinuous scale for genetic divergence. We also removed any reference to the concept of a gradually increasing distance in the text.

1.4 Given the tight word limit you are given; it might be better to tell the reader upfront that your parental lines differed in their initial ploidies and that that is in fact what determined the vastly different outcomes in fertility you observed. In the end, genetic distance played only a minor role in determining the chances for fertility discovery in your study, while ploidy overrides all other effects.

Answer 1.4: We agree with this statement. We have now reduced the length of the discussion regarding the triploid hybrids and removed this aspect from our discussion. We discuss upfront what they are and where they come from so, we do not have to discuss about this issue any longer that a few lines. However, to be consistent with the logical flow of the paper and because there was no reason to measure ploidy before addressing the issue of fertility, we prefer to keep this part where it is now.

1.5 Line 22: “which can hinder their potential as new species or populations.” Consider rephrasing. It is not clear what you mean with “potential as new species”.

Answer 1.5: We replaced this with: “*which can prevent their maintenance as independent populations, thus hindering their speciation potential.*”

1.6 Line 19-31: I think the evolutionary logic of this argument here is not quite right. So because some species have access to both sexual and sexual reproduction, their hybrids

can restore fertility via genome doubling. The way it is phrased here, it sounds as if the reverse argument would be that obligatory sexual species do not have access to whole-genome doubling (which is of course not the case as you say at the end of the abstract). As a result, this reads a bit as if evolution in non-obligatory sexual species is intentionally working towards hybrid speciation (while really hybrid speciation is more a by-product of genome doubling).

Answer 1.6: We changed the introductory paragraph according to the reviewer's comments 1.6 and 1.7. See changes in the next answer.

1.7 Also, I think the comparison to obligatory sexual species in this context does not really work since in obligatory sexual species, backcrossing with parents usually leads to introgression, but not to hybrid speciation. Instead, to make this comparison valid, why not introduce the concept of homoploid hybrid speciation here and say that in obligatory sexual species homoploid hybrid speciation is rare and usually requires other forms of isolation (e.g. ecological or geographic) of the hybrids from the parents, in addition to restoration of fertility in hybrids through recombination. I think comparing your case with homoploid hybrid speciation sets the stage better for you to show that whole genome duplication in non-obligatory sexual species provides another way for hybrid speciation to work.

Answer 1.7: We changed the introductory paragraph according to the reviewer's comment. We put more emphasis on the fact that hybrid speciation is more difficult in sexual species because of the ongoing gene-flow with parental species and that asexual phases might provide opportunities that would facilitate hybrid speciation. (line 42-49). We do not talk about homoploid hybrid speciation however because our results show that change in chromosome numbers may be important and homoploid speciation regards speciation without such changes.

"Because of this gene-flow, this process most often leads to the formation of introgressed species⁸ rather than hybrid species⁹. In this context, the formation of hybrid species may necessitate other means of isolation from both parental species, which may include geographic or ecological isolation while the recovery of fertility through recombination takes place¹⁰. Some organisms, however, have access to both sexual and asexual reproduction. In these species, if sexual encounters are rare, periods of asexual reproduction might provide hybrids with alternative mechanisms for fertility recovery, which could facilitate hybrid speciation."

1.8 Line 44-55: I think it would be fair to mention early on here, that the genetic distances of the crosses you made do not cover the 15M years of divergence evenly. Instead, the majority of your crosses cluster around very small to small distances (0-4%), then there is a big gap, and only then come the large interspecific Sp x Sc distances (15%). This fact

becomes masked in your categories of VL, L, M and H, which may mislead readers into believing the distribution of crosses is continuous along the genetic distance axis. This is especially evident in Figure 1B and 1E where the bar plots give the impression of equal intercepts of genetic distances between the four crosses.

Answer 1.8: We agree that this can be misleading in Figure 1B because the scale looks continuous, but the species/strains are not proportionally spaced on the scale. We therefore changed this figure to show that there is break in the distance (see the changes in figure 1B in comment 1.3). However, for figure 1E, these are categorical plots so the spacing on the x-axis has no meaning.

1.9. Line 52: Consider rephrasing “in terms of DNA sequences”.

Answer 1.9: We replaced "Including this sister species extends the genetic divergence of the crosses to 15% in terms of DNA sequences." With (line 70-71):

“Including this sister species extends nucleotide divergence between parental strains up to 15%”

1.10. Line 69: Your results here are not in conflict with references 13, 14 and 15. These studies have all shown that a small fraction of the hybrid offspring of inter-specific yeast crosses have higher fitness than the parents and therefore may harbor some adaptive potential, but these transgressive hybrids are not representative of the average fitness of the cross.

Answer 1.10: We agree that our results are not in conflict with these references we thus removed this sentence.

1.11 Line 71-72: “Hybrids could also have access to more deleterious genomic changes”. I think this is a strong understatement. Most hybrids from divergent crosses will on average be less fit than the parental pure-bred crosses as hundreds of studies on hybrid break-down or outbreeding depression have shown, in many different (mainly eukaryote) species.

Answer 1.11: We agree with the reviewer we thus removed this sentence.

1.12 Line 78-79: The loss of the ability to sporulate is a rather common outcome in yeast strains subjected to experimental evolution (and is probably why laboratory strains like S288c are bad sporulators). That doesn't take away from your findings but should probably be mentioned here.

Answer 1.12: We could not find any reports of loss of sporulation in the literature for other mutation accumulation experiments as the one we performed. Hill and Otto 2007 reported

that the sporulation rate declined by 8% on average and concluded that most mutation affecting sporulation rate are deleterious. Zeyl 2005 also reported declines in sporulation rates from both a bad (10% ancestral sporulation rate) and good (60% ancestral rate) sporulator. We think that our results are different in that we show complete loss of sporulation ability (>90% ancestral sporulation rates). It is also to note that these experiments were made using non-hybrid yeast backgrounds, in which case sporulation ability maybe lost at a slower rate. However, we thank Dr Stelkens for this suggestion and we now cite previous studies that looked at this.

Added to the discussion (line 278-288):

“One of the most striking changes to fertility we observed is a complete loss of the ability to sporulate within thirty generations, effectively reducing fertility to 0. While decrease in sporulation efficiency was observed in similar experiments performed on homozygous strains^{39,40}, such complete loss of the ability to sporulate in strains that initially were able to sporulate at high efficiencies was, to our knowledge, not reported before. Our results show that a greater genetic distance between the founding parents lead to higher probability of losing the ability to sporulate, suggesting that genome instability or genetic incompatibilities could cause this decrease of fertility. This, added to the results indicating that those strains also lost some or all of their mitochondrial DNA and growth defects on non fermentable sources, indicate that genetic incompatibilities or instabilities involving the mitochondrial genome may be responsible for these fertility losses.”

1.13 Line 79: I don't find the overall negative relationship between sporulation efficiency and genetic distance of the cross surprising. That is what we would expect from more and more divergent genomes, i.e. increasing sterility with increasing distance. The most interesting and unexpected element here is that the blue cross (low divergence) has much lower survival than the two more divergent crosses (Figure 1D). I think that is what should be emphasized over all other results here.

Answer 1.13: What is surprising here is that all of those strains were initially able to sporulate right after hybridization (at >90% rate) so sporulation ability is not affected by genetic distance itself, but genetic distance has an effect on the evolution of sporulation through time. The strains were not initially sterile but became sterile while evolving. Also, the rapid timeframe of the losses would coincide with fixation of mtDNA haplotypes. Our PCR on mtDNA loci (Supplementary figure 4) suggests that mitochondrial elements were lost. The increasing probabilities of such events with genetic distance could suggest that genetic incompatibilities (mito-mito or cytonuclear) are at play here (see discussion additions above).

1.14 Figure 1E. The black and white in the stacked bars is not explained in the figure legend. What am I looking at here? This is the first time it is mentioned that two different parental lines were used.

Answer 1.14: The two different colors were indeed representing the *SpB* parents used in the crosses. We tried to make it clearer at the beginning in the text that we used two different strains. However, we did remove them because there were no significant differences in the proportions of strains that lost sporulation between parents. This will lessen possible confusion.

Introduction changes (line 71-73):

“We mated two SpB strains to two other SpB strains and to two strains of the diverged lineages and species, producing 4 different types of crosses in duplicates that we classified in terms of divergence [...]”

1.15 Figure S2: Similarly here, it is not immediately obvious to me what set 1 and set 2 are since this is not explained in the figure legend.

Answer 1.15: Set 1 and Set 2 were indeed representing the different parental strains. We completely changed the legend in the figure, so each individual cross is represented.

1.16 Line 83: Are there alternative explanations for bad sporulation capacity (other than loss of mitochondria) that should be mentioned here?

Answer 1.16: There are more than 200 nuclear genes that when knocked down lead to an absence of sporulation. Loss of function mutations in those genes could therefore lead to complete loss of sporulation. However, the frequency of the loss of sporulation events (between 2% and 22% of lines) and the timing of the events (within the first passage following hybridization) suggest that mutations would probably not be the main factor at play here. However, there is some rare case where colonies are unable to sporulate but still have their mitochondrial DNA which could be explained by nuclear genome mutations or mitochondrial point mutations. We now changed the text to (lines 132-137):

*“There are also rare cases in which lines lost their sporulation ability while both mitochondrial markers were detected (Supplementary Figure 4), suggesting that the loss of mtDNA integrity is not the only cause of sporulation inability. As the Saccharomyces Genome Database (SGD) reports that there are more than 200 genes that, when knocked out in *S. cerevisiae*, lead to an absence of sporulation²⁵, any loss of function mutation happening in one of those genes could explain this phenotype.”*

1.17 Figure 2B: It would help readability of the figure if you made clear on the actual figure that graph 1-4 are a result of mitosis, and graph 5 is the result of meiosis.

Answer 1.17: We have changed the order of the panels to regroup panel 1-4 (mitotic) together and put graph 5 (meiotic) alone on the third column and identified them as mitotic or meiotic.

1.18 Line 92: Please explain if these are ITCs from all 4 cross types (VL, L, M and H). Given Figure S6, it looks that all ITCs are derived from SpB x SpC crosses? I also believe that the figure reference should be to Fig. 2B not C.

Answer 1.18: As suggested by fig S6, all the ITC were made from L crosses only. This is also mentioned in the method section. We added a mention to the concerned lines in the main text (lines 158-161). The figure reference has also been changed to 2B.

“As a point of comparison for fertility restoration with sexual reproduction, we performed 12 meiotic generations of intra-tetrad crosses (ITC) in randomly chosen diploid Ldiv lines and calculated FRS, all of which were positive (Fig. 2C, Supplementary Figure 6)”

1.19 Line 96: I am not sure I understand this test. You autodiploidized (selfed) these strains to make them fully homozygous. So this tests for the presence of any intra-chromosomal (homologous) incompatibilities between the two divergent genomes present in the hybrid. At the same time this controls for potential effects of antirecombination, preventing crossing over and leading to the missegregation of chromosomes during meiosis, which can result in aneuploidies. However, autodiploidization does not exclude potential negative epistatic effects between non-homologous chromosomes, which may also lead to low spore viability. Perhaps I am misunderstanding something here but this may need some more explanation.

Answer 1.19: We agree with the reviewer that this sentence was not clear. Autodiploidization does not exclude the possibility of low fertility resulting of negative epistatic interactions. The fact that we could only select viable spores for this experiment eliminates the possibility of

seeing such interactions, at least the most deleterious ones. Since most of the infertility in *Saccharomyces* has been reported to be due to anti-recombination between homeologous chromosomes, we thought this would be a good control for the potential of fertility recovery via LOH events. However, in figure 2C we can see that 2 hybrids remain with lower fertilities (around 50%) even after autodiploidization. Slight changes were made to the phrasing (Line 164-168):

“To make sure that low spore viability was not due to the intrinsic inability of strains to produce viable spores, due to dominant mutations for instance, we performed autodiploidization on a random set of 16 haploid spores from the Ldiv and Mdiv crosses and this, at the three timepoints. In most cases, fertility was restored to more than 85% upon selfing (Fig. 2C), showing that infertility mostly derives from the presence of two divergent genomes in the same cell”

1.20 Line 101-105: Please present some statistical evidence and add a figure for this result: “...we observed cases of spectacular fertility recovery in seven mitotic lines (4 Ldiv lines, 2 Mdiv lines and 1 Hdiv line), which displayed FRS close to that of ITC lines.” If these are the seven lines shown in the histogram in Figure 2b (the outlier bins) it might be a good idea to highlight this in the histogram.

Answer 1.20: We put red arrows over the concerned bins to better identify where the strains are located in the histograms of FRS. We also tested the differences in fertility between T_{ini} and T_{end} for each line with a Fisher exact test on the proportion of viable spores. As expected, the difference in the fertility between T_{ini} and T_{end} were all statistically significant for the tetraploid lines. The information about which lines is significantly different between T_{ini} and T_{end} is now found in Figure 4. We added some changes to the text that mention results of the analyses (lines 169-180).

*“Although no general trend towards the recovery of spore viability was observed during the experiment, 23 individual lines presented statistically significant difference in their fraction of viable spores between T_{ini} and T_{end} (Fisher’s exact test, *fdr* corrected, Supplementary Table 6). Eleven of those 23 lines (5 VLdiv, 5 Ldiv and 1 Mdiv) showed a decreased fertility and the remaining 12 (7 Ldiv, 4 Mdiv and 1 Hdiv) showed improvement. Among the lines that showed improvement, seven (4 Ldiv, 2 Mdiv and 1 Hdiv) displayed spectacular recovery in spore viability within 352 mitotic generations: their FRS values were close to that of ITC lines (Fig. 2B), and their spore viability was similar to what is typically observed for their non-hybrid diploid parents¹⁷. One line presented a low FRS value, which is explained by the fact that its fertility returned to its initial value by T_{end} (Fig. 2B). As infertility of *Saccharomyces* hybrids is mainly due to anti-recombination caused by the mismatch*

repair machinery acting on homeologous chromosome pairs, leading to chromosome missegregation^{28,29} [...]”

1.21 Line 112: It is not clear (to the non-yeast scientist) why autodiploidization is less likely an explanation than autopolyplodization here.

Answer 1.21: We added a part explaining both that yeast sporulates under specific environmental conditions and that those conditions were not supposed to be met in the context of our experiment (not allowing enough time for yeast to deplete the medium) (lines 192-199). We also added a part about the HO locus deletion, which prevented real autodiploidization (lines 242-244).

Lines 192-199:

“The last potential mechanism would be that strains could have sporulated during the experiment and spores of opposite mating types could have mated. This would be the equivalent of our ITC lines where sometimes a single cross between two spores can bring fertility back to high values (Supplementary Figure 6). This third

option is very unlikely because sporulation happens under very specific environmental conditions, principally nitrogen starvation. The frequency of streaking to fresh media during the experiment would prevent such depletion to happen. In addition, this scenario would often lead to spores that are aneuploid, making fertility recovery unlikely even after mating.”

Lines 242-244:

“In this experiment, mating type switching may not occur using the standard process because the necessary HO gene was deleted”

1.22 Figure 3 C: the variation along the y-axis indicates that ploidy was not an exact measure (i.e. ploidy did not come out exactly as 2N, 3N or 4N). You don't mention the possibility that some of these strains may have aneuploid chromosome content at all. Is it possible that you have seen many different aneuploidies come and go in the genomes over the course of experimental evolution (as e.g. seen in Yona et al. PNAS 2012)? It would be good to see a discussion of this somewhere in the paper.

Answer 1.22: Indeed, we observed several aneuploidies in our hybrid sequenced genomes. We added this information in the legend of Figure 3A:

“black circles indicate the highest cell count whose variability suggest that aneuploidy is prevalent in hybrid lines.”

and in the text (lines 263-270)

*“However, we cannot exclude that damage to the MAT loci occurring by the loss of chromosome III could be undetectable because this chromosome could have been regained following tetraploid formation. Chromosome III was indeed found to be the most unstable chromosome among *S. cerevisiae* diploids, triploids and *S. cerevisiae*-*S. bayanus* hybrids³⁵. Furthermore, transient dynamics of aneuploidy were repeatedly observed in yeast laboratory evolution experiments under stressful conditions^{37,38}. Such successive aneuploidies could happen in our hybrids lines because aneuploidy is prevalent (Fig. 3C, Supplementary Figure 18) and tetraploids are notorious for showing genome instability³⁶.”*

The supplementary figure S18 show some cases of aneuploidies:

“Supplementary Figure 18. The loss of chromosome III is not the molecular mechanism leading to whole genome doubling. Sequencing read depth for bins of 10kb on the 16 chromosomes of the 8 tetraploid lines at T_{ini} (the left panel) and T_{end} (the right panel). The red line represents the average sequencing read depth of the whole genome. Several aneuploidies are detected in almost all hybrids at T_{ini} and T_{end} .”

1.23 Line 136: There appears to be many more than 6 lines showing more than 70% fertility after evolution in Figure 4. The remaining lines, outside of the 6 that you mentioned, were all in the VL divergence cross. These lines all remained diploid. I think it would be important to mention this because the way it currently reads is that the only possibility of maintaining >70% fertility is to become tetraploid.

Answer 1.23: It might not have been clear in the text, but as the VL lines represent within group crosses, they were expected to show high fertilities and act as “control lines”. We also expected this fertility to remain fairly the same, or even decrease, as there was little place for improvement of fertility as the heterozygosity could probably only increase in those strains following mutation accumulation.

We did obtain a low fertility *SpB*×*SpB* cross (VL2), which is not a surprise because *SpB* is fairly diverse both in genomic sequence and karyotypes. Overall, they did behave like the other crosses, but were the only ones that did not show tetraploids. There also seem to be strains with large decrease of fertility in this group. In order to avoid possible confusion, we removed any reference to lines showing more than a percentage of recovery and expressed ourselves either about significant amelioration over time or comparison with non-hybrid backgrounds fertilities we observed in previous work (Charron *et al.* 2014 or Leducq *et al.* 2016)

1.24 Also, in Figure 4 in the VL crosses it appears that there was a difference between the two independent crosses (dotted and full lines). Is this difference significant and can it be explained?

Answer 1.24: This difference is statistically significant. As the two crosses were made with two independent sets of strains, this difference is not unexpected. The median fertility of intra lineage crosses for *SpB* is about 50% (figure S3) with crosses as high as 80% and as low as 35-40% (Charron *et al.* 2014, molecular ecology, Leducq *et al.* 2016). *SpB* is a group which harbours the most among strain genetic and karyotypic variation which probably explains the difference between the two L_{div} crosses. We now mention this in the text to make things clearer (lines 116-120):

“However, there were significant differences between the biological replicates (one way ANOVA $F(7,552)= 171.8$) in the VL_{div} (averages of 47.2% and 73.4, $P < 0.01$, Tukey HSD) and M_{div} crosses (averages of 36.4% and 18.1%, $P < 0.01$, Tukey HSD). These differences are probably due to strain specific genetic variation or even genomic architecture leading to variable levels of postzygotic isolation¹⁹”

1.25 Lastly, in Figure 4 it appears that in the M crosses there was a line that was tetraploid at T_{mid} but became diploid at T_{end} . This would be worth mentioning.

Answer 1.25: We added a short discussion about the M cross (M1_40) that was tetraploid at T_{mid} and became diploid at T_{end} in the text (lines 233-236):

“One of the two tetraploid M1 lines (M1_40) returned to diploidy at T_{end} but this is more likely to be due to segregating ploidy and colony heterogeneity at T_{mid} rather than return to diploidy (for more details see Supplementary text).”

We also added a supplementary material’s section about this M1 line “Return to diploidy after tetraploidy” (lines 64-71):

“Return to diploidy after tetraploidy

These analyses allowed us to examine the genome of the M1 tetraploid (M1_40) whose ploidy and fertility increased at T_{mid} and then decreased at T_{end} . This result

could be explained by the presence of heterogenous colonies during our evolution experiment. The isolated colony at Tmid might segregating diploid and tetraploid cells. One type may have fixed in the glycerol stock and be lost in the next round of propagation, explaining why we observe only tetraploids at Tmid. However, the colony isolated during the subsequent passage was a diploid one explaining why we observe only diploids in subsequent passages.

1.26 Line 148: I don't follow the argument in this sentence: "Mitotic fertility recovery contrasts with our results and that of others showing that sexual reproduction leads to incremental and rapid recovery of fertility". It is unclear what argument is being made here. As I understand it, this study showed that mitotic fertility recovery is possible, so therefore how does it contrast with your results?

Answer 1.26: we agree with the reviewer that this part of the text was not clear. We completely revamped this section of the text (Lines 272 to 319) in order to better connect our results with the conceptual figure presented in Figure 1A (upon suggestion by reviewer #2). Concerning the point raised by the reviewer, we added the following sentence (lines 312-321):

"Finally, a more gradual form of statistically significant recovery was also observed in about six lines which were not subjected to genome duplication. For four of those lines, fertility almost doubled compared to their ancestral lines. The other two have improvement of 3 and almost 5 times their initial fertilities. As mitotic recombination often leads to gene conversion events spanning tens of thousands of base pairs⁴², accumulation of a large number of mitotic loss of heterozygosity events might allow such recoveries. More in depth genomic analyses will therefore be needed to understand the basis of these recoveries. This would suggest that the contribution of mitotic recombination to the recovery of fertility in hybrid lineages could be a slow process compared to meiotic recombination and whole-genome duplication."

1.27 Line 160-163: Consider specifying here that the "multicellular eukaryotes" you used to support your statement about fertility restoration without sex are in fact plants. It would be good to see a discussion here on whether you think WGD would also be a functional mechanism for the restoration of fertility in animals as you mention in the abstract. I think it would be difficult to find many supporting references for fertility recovery without sex in animal hybrid crosses.

Answer 1.27: We added a sentence following this part which add nuances about plants and animals and the plausibility of this mechanism (lines 353-357):

“While polyploidy has been shown to be prevalent in plants⁴⁶, polyploidy is not common in animals^{47,48}. However, the animal lineages containing stable polyploids species often have access to asexual reproductive strategies such as parthenogenesis which improve tolerance to polyploidy and could enable hybrid fertility restoration without sex⁴⁹.”

1.28 Line 330: Replace “cross” with “crosses”.

Answer 1.28: Replacement was done

1.29 Line 345: Should this be Table S2?

Answer 1.29: This should indeed be table S2. However, this table is now Supplementary Table 8 and was changed in the text accordingly.

Reviewer #2 (Remarks to the Author):

2.1 The study by Charron et al. examines neutral fertility recovery among hybrid strains of varying divergence levels passed through repeated mitotic bottlenecks for ~800 generations. Interestingly, although the initial probability of successful sporulation was negatively correlated with parental divergence, the recovery of fertility (measured through spore viability assays) evolved neutrally – strains were generally equally likely to gain or lose fertility. The exception was ploidy-variant strains: triploid strains were more likely to have low fertility, while tetraploid strains were more likely to have high fertility. I generally found this to be an enjoyable read of an interesting and novel story about the interplay between strain divergence and ploidy on neutral recovery of fertility following hybridization.

Answer 2.1: We thank the reviewer for his favorable review.

2.2 Based on the format and brevity of the text, it seems that this manuscript was previously submit to a journal with more strict word limits, and not revised prior to submission here. I appreciate that the format should not need to be changed prior to re-review here (and indeed strongly believe that initial review at all journals should accept manuscripts in any format the authors choose). Many of my concerns however are as a direct result of (my perceived) previously-imposed brevity and my overarching request is that the authors better analyze and discuss the intricacies of the interesting data they have collected.

Answer 2.2: We agree that the clarity of the results reported here suffered from the brevity of the text. We tried to expand on the methods, results and their analyses to minimize the potential confusing parts that the previous version suffered from.

Major questions

2.3. A nice component of this work is that the experiments were carried out on two independent crosses for each divergence class, and it is of general interest to know how strain background differences influences the results. Hence, it was surprising to see that the authors did not generally present the data or provide a narrative about the similarities or differences in results from different crosses of the same divergence class. Since there is internal structure of strain relatedness, it seems critical that the data not be collapsed together without first statistically demonstrating that there were no significant differences between crosses of the same divergence class (e.g., in sporulation capacity, survival, ancestral spore viability, etc.) and/or taking into account the relatedness structure in analyses. The data in figure 4 (initial fertility) is plotted (but not discussed) by background and there does seem to be a background effect for VL and M strains but not for L?

Answer 2.3: Following the suggestion of the reviewer, we tried to better discuss the difference that could be found between replicate lines of the evolution. We averaged the values when they were not statistically different and avoided to collapse the significantly different values. We changed figure 1D to collapse only the crosses that had significantly different survival. We collapsed the figure 1E results because there was no significant difference between the cross mad with different parental strains.

Comparison of initial fertilities (lines 116-120):

“However, there were significant differences between the biological replicates (one way ANOVA $F(7,552)= 171.8$) in the VLdiv (averages of 47.2% and 73.4, $P < 0.01$, Tukey HSD) and Mdiv crosses (averages of 36.4% and 18.1%, $P < 0.01$, Tukey HSD). These differences are probably due to strain specific genetic variation or even genomic architecture leading to variable levels of postzygotic isolation¹⁹.”

Discussion point about variation (lines 323-330):

It is important to note that our results also show that there is some variation for fertility and ploidy 1) within the colonies that were replicated and frozen, 2) among lines within a cross and 3) between biological replicates of the same divergence category. This variation could be due to the interactions of the parental genetic backgrounds, generating instability in the hybrid lines, and their elevated instability that starts in the zygotes after the initial matings. Therefore, taking into account the initial genetic background of experimentally evolved hybrids and considering multiple independent hybridization events is crucial for the understanding of their possible evolutionary fates.”

Log-rank test P-values

	VL1	VL2	L1	L2	M1	M2	H1
VL2	0.42						
L1	< 0.01	< 0.01					
L2	< 0.01	< 0.01	0.63				
M1	< 0.05	< 0.05	0.15	0.35			
M2	0.24	0.75	< 0.01	< 0.01	< 0.05		
H1	< 0.01	< 0.01	0.81	0.75	0.35	< 0.01	
H2	< 0.01	< 0.01	0.42	0.75	0.47	< 0.01	0.60

2.4. The survival (extinction) data is interesting, particularly in contrast with the heterosis has been previously reported with large populations. I found the language to be a bit misleading (L69) since these results aren't really in contrast, and as stated on L71 it really just indicates that the hybrid populations are incredibly diverse, and have access to both more deleterious and more beneficial mutations. In the vein of presenting all the data, it would be good to have a table that indicates which strains went extinct (was it equally likely from both backgrounds?). I'm wondering whether the authors explored intra-population viability/probability in survival– if you streak multiple colonies from the same population that went extinct, how often do you observe extinction vs. survival?

Answer 2.4: There is now a table containing the strain information regarding the passage at which they were lost in the source data file supplementary. The differences in survival between individual crosses are now shown in Figure 1. As suggested by the reviewer, we devised a small-scale experiment where we got 4 strains that went extinct (from L and H crosses as those were the one we lost the most strains of) and 4 control strains that survived until the end of the experiment.

For each of those strains, we isolated 24 clones from the glycerol right before extinction (each extinct strains were paired with a control of the same genetic background taken at the same timepoint). We then propagated those clones for 4 passages using the exact same procedure as in the evolution experiment. Surprisingly, we lost only 1 strain and it was a control.

We think that there might have been some heterogeneity in the genotypes of the colonies. This could have led to discrepancies of the phenotypes of the colonies during the experiment as compared to the glycerol stocks because the preculture of the glycerol could have acted as a competition culture in which the fittest/stable genotype would take over the genetically unstable ones.

We do mention the possible heterogeneity of the colonies in a supplementary text, but we do not mention the new small-scale experiment.

2.5. Aneuploidy is not discussed but is potentially prevalent. Is everything in Figure C3 euploid? The variation in Figure 3A would suggest to me that aneuploidy is prevalent in VL_T_{ini} and H_T_{ini} strains (as well as potentially all of the L_T_{ini} strains), but if the GBS data shows euploidy then perhaps the mean or median black squares in Figure 3A (what this is should be specified in the figure legend) is misleading.

Answer 2.5: Indeed, aneuploidy is prevalent in our different crosses and this is observable by GBS and whole genome sequencing data. We added this in the text (lines 263-270) and supplementary Figure S18.

“However, we cannot exclude that damage to the MAT loci occurring by the loss of chromosome III could be undetectable because this chromosome could have been regained following tetraploid formation. Chromosome III was indeed found to be the most unstable chromosome among S. cerevisiae diploids, triploids and S. cerevisiae-S. bayanus hybrids³⁵. Furthermore, transient dynamics of aneuploidy were repeatedly observed in yeast laboratory evolution experiments under stressful conditions^{37,38}. Such successive aneuploidies could happen in our hybrids lines because aneuploidy is prevalent (Fig. 3C, Supplementary Figure 18) and tetraploids are notorious for showing genome instability³⁶.”

“Supplementary Figure 18. The loss of chromosome III is not the molecular mechanism leading to whole genome doubling. Sequencing read depth for bins of 10kb on the 16 chromosomes of the 8 tetraploid lines at T_{ini} (the left panel) and T_{end} (the right panel). The red line represents the average sequencing read depth of the whole genome. Several aneuploidies are detected in almost all hybrids at T_{ini} and T_{end} .”

The black squares in Fig. 3A as in 3B represent the fluorescence value of the highest cell count, we now specified that in the legend of Fig. 3A. The observed variation indeed suggests that aneuploidy is prevalent.

“... black circles indicate the highest cell count whose variability suggest that aneuploidy is prevalent in hybrid lines.”

2.6. The result that a small fraction of diploid cells in the parental stock led to 50% triploidy in the initial crosses in only some backgrounds seems to suggest a conditional mating

advantage for diploids that is background/context dependent. It would be nice to see this fleshed out a bit. Is it just coincidental that both crosses used for the evolution experiments (L1 and L2) exhibit the high prevalence of triploidy when the others do not?

Answer 2.6: The proportion of diploid cells in the parental haploid stock appears to be variable and is consistent with stochastic switching at low frequency. We repeated the experiment of ploidy analysis of the parental stock two times, the first time no diploid was identified among 94 tested isolated colony. However, the second time 5 among 94 were identified as diploid. This suggests that the duplication of haploid cells in the parental stock is a stochastic event. Moreover, the frequency of triploids in L1 and L2 crosses is variable and may vary from 1% to 57% as shown in the table S5. These results suggest that the proportion of triploids in hybrids might depend on the initial proportion of diploid parents (pseudohaploid) in the preculture, which itself is stochastic. The frequency of diploids in the preculture used for the hybridization might also depend on at which phase of cell growth it appears to be more or less widespread in the cell culture. The reason why some haploid strains diploidize remain unknown and need more investigations. Some genetic or transcriptomic variations that affect some key genes involved in cell cycle regulatory pathways could be on the origin of this process. The absence of triploids in some other L crosses indicate that diploidization in parental strains is background dependent. Indeed, phenotypic, genomic and karyotypic diversity have been observed among strains within *SpB* and *SpC* lineages (Leducq et al., 2016).

2.7. A suggested discussion topic/framework is to tie the results back to Figure 1A – you have nicely set up model scenarios for fertility recovery and could easily describe your results in the context of this framework.

Answer 2.7: We agree with the reviewer that we didn't make use of the conceptual figure. We therefore added a discussion paragraph (lines 278 to 321) returning to the possible scenarios and if they were observed in the context of our experiment. We thank the reviewer for his suggestion.

“One of the most striking changes to fertility we observed is a complete loss of the ability to sporulate within thirty generations, effectively reducing fertility to 0. While decrease in sporulation efficiency was observed in similar experiments performed on homozygous strains^{39,40}, such complete loss of the ability to sporulate in strains that initially were able to sporulate at high efficiencies was, to our knowledge, not reported before. Our results show that a greater genetic distance between the founding parents lead to higher probability of losing the ability to sporulate, suggesting that genome instability or genetic incompatibilities could cause this decrease of fertility. This, added to the results indicating that those strains also lost

some or all of their mitochondrial DNA and growth defects on non fermentable sources, indicate that genetic incompatibilities or instabilities involving the mitochondrial genome may be responsible for these fertility losses.

We find that reproductive isolation as assessed by spore viability does not have a global directional fate during the mitotic propagation of hybrid lines. The different scenarios presented in figure 1A were almost all observed. Overall, spore viability of hybrids evolved like a neutrally evolving quantitative trait, with incremental gains and losses, with no overall particular direction. However, we did observe spectacular punctuated improvements in spore viability for some lineages. From the 23 lines that had statistically significant differences between the initial hybridization and the end of the experiment, fertility decreased for 11 of them. Reductions in spore viability was observed in all crosses except the Hdiv crosses, which were already completely infertile at the beginning of the evolution and could not be reduced further. One explanation for such decrease includes the accumulation of genomic rearrangements, which would lead to incorrect segregation of chromosomes during meiosis to various degrees depending on the extent of the rearrangements^{14,19}. The segregation of recessive lethal alleles following a de novo mutation could also be implicated and would lead to strains exhibiting halved fertilities, which were rarely observed, but more complex patterns of fertility decrease are also possible given the potential for genetic interactions in these heterogeneous genetic backgrounds.

Punctual and almost complete restoration of fertility was observed at low frequency in all but the lowest parental divergence crosses. We show that these lines experienced a genome duplication, most likely caused by the doubling of all chromosomes. It was shown before that artificially induced chromosome doubling in infertile hybrids between *S. paradoxus* and *S. cerevisiae* hybrids could restore fertility⁴¹. Our results show that this happens spontaneously, without the need for natural selection and in between species that can naturally hybridize. Finally, a more gradual form of statistically significant recovery was also observed in about six lines which were not subjected to genome duplication. For four of those lines, fertility almost doubled compared to their ancestral lines. The other two have improvement of 3 and almost 5 times their initial fertilities. As mitotic recombination often leads to gene conversion events spanning tens of thousands of base pairs⁴², accumulation

of a large number of mitotic loss of heterozygosity events might allow such recoveries. More in depth genomic analyses will therefore be needed to understand the basis of these recoveries. This would suggest that the contribution of mitotic recombination to the recovery of fertility in hybrid lineages could be a slow process compared to meiotic recombination and whole-genome duplication.”

Minor concerns

2.8 L49 Genetic divergence between SpA x SpB and SpC x SpB is presented in Figure 1B and could be referenced here (until I saw this figure I also wondered about divergence among the different paradoxus lineages). How much divergence is there among different strains of the same lineage?

Answer 2.8: A recent study by Eberlein et al., 2019 gave an overview about divergence among the different paradoxus lineages and among different strains of the same lineage. We added this reference and a supplementary table S2 presenting this information.

“Supplementary Table 2.

Genetic divergence among the different paradoxus lineages and among different strains of the same lineage (Eberlein et al., 2019).

Reference	Compared to	% Mean nucleotide divergence
SpB	SpB	0.424
SpB	SpC	2.237
SpB	SpA	3.748
SpC	SpC	0.288
SpC	SpA	3.432
SpA	SpA	0.101

“

2.9 L49 “up to 60% reduction in fertility in hybrids” -> this is confusing. Reduced fertility in which hybrids and relative to what?

Answer 2.9: We added information to the sentence in order to clarify it (lines 66-68):

“The S. paradoxus lineages (SpA, SpB and SpC)¹⁶ are incipient species that exhibit up to 4% of genetic divergence (SpA-SpB)^{17,18} and up to 60% of reduction in hybrid fertility compared to within lineage crosses¹⁹”

2.10 L53-58: I found the section in the text that described the crossing scheme quite confusing and it took me awhile to reconcile Figure 1B and Table S2. It would be helpful to

keep the nomenclature consistent throughout. The abbreviations S.pa1 and S.pa2 are unique to this figure panel, I suggest using SpA, SpB and SpC here instead. There are also extra strains listed in Table S2 that were not used in the evolution experiment. Of very minor note, the MSH604 x LL12_028 cross listed in Table S2 should be moved above the UWOPS-91_202 lines.

Answer 2.10:

-We changed figure 1B as the reviewer suggested by using *SpB1*, *SpB3*, *SpC1*, *SpA1* and *S.ce1* for the first cross repetition and *SpB2*, *SpB4*, *SpC2*, *SpA2* and *S.ce2* for the cross repetition 2 instead of S.pa1 and S.pa2.

-Table S2 (S3 after revision) Lists all crosses made for this study. The “extra strains” listed in Table S2 were indeed not used in the evolution experiment but were used for additional experiments presented in supplementary figures to evaluate if triploidy is observed in crosses between *SpC* and other lineages and species (Fig. S11) and in crosses between different *SpB* and *SpC* strains (Fig. S12).

-We moved MSH604 x LL12_028 (VL_{div1}) above the UWOPS-91_202 x LL12_021 (VL_{div2}) in the table S3.

“Supplementary Table 3.

List of crosses made for this study

Cross	a strain	a strain resistance	α strain	α strain resistance
VL _{div1}	MSH604	NAT	LL12_028	G418
L _{div1}	MSH604	NAT	LL11_004	G418
M _{div1}	MSH604	NAT	YPS644	G418
H _{div1}	MSH604	NAT	LL13_040	G418
VL _{div2}	UWOPS_91_202	G418	LL12_021	NAT
L _{div2}	UWOPS_91_202	G418	LL11_009	NAT

M_{div2}	UWOPS_91_202	G418	YPS744	NAT
H_{div2}	UWOPS_91_202	G418	LL13_054	NAT
L_{div3}	UWOPS_91_202	NAT	MSH587	HYG
L_{div4}	YPS484	NAT	YPS667	HYG
L_{div5}	yHKS226	HYG	yHKS225	NAT
L_{div6}	LL12_021	NAT	LL12_019	HYG
SpCxSpA-1	YPS744	NAT	LL11_004	G418
SpCxSpA-2	YPS644	G418	LL11_009	NAT
SpCxSpC-1	LL11_009	NAT	LL11_004	G418
SpCxScer-1	LL13_054	NAT	LL11_004	G418
SpCxScer-2	LL13_040	G418	LL11_009	NAT

2.11 Figure 3: What is the X axis (“fluorescence”) scale?

Answer 2.11: The X axis in figure 3 A represent fluorescence in arbitrary units, we added (a.u) in the X axis label of Figure 3 A, Figure S9, Figure S11 and Figure S12.

Example of Figure 3 A:

2.12 Figure 4: There seems to be a drop in fertility for T_{mid}, at least for M1 strains and possibly L1 strains. Is there a biological or methodological explanation for this? If methodological, does this impact other results?

Answer 2.12: We agree with the reviewer that the fertility seem to drop at T_{mid} for those crosses and you can clearly see an increase in the proportion of 0 spore tetrads in Fig S7 at that time point. For logistics reasons, T_{ini} and T_{end} strains from the same cross types (examples: L1 and L2 lines) were sporulated at the same time while T_{mid} strains were sporulated later. We also suspect that the change in yeast extract could have played a role in this as the cells grow slower on it.

The first batches of dissections were done for T_{ini} and T_{end} (L, M and H) between November 2017 and January 2018. The T_{mid} batches were done between March 2018 and April 2018. We verified orders of Yeast Extract and there was one in February 2018, which would further comfort that the dissections were done on different batches of media using the two different lots of yeast extract. This could have made a difference in the detection of smaller sized colonies and lead us to underestimate the fertility of the T_{mid} lines for the L and M crosses. We think this should not affect our results as the main comparison we're making is between T_{ini} and T_{end}, which were done using the same media.

We mention the effect of yeast extract in the material and method section (ade2-Δ colony coloration phenotype, lines 598-606):

“Although we used the ade2-Δ marker as a visual aid to track loss of mitochondrial DNA, we still passaged strains that seem to have lost mitochondrial DNA. During the experiment, some colonies from all crosses suddenly turned whitish or light orange. This paler coloration did correlate with the absence of sporulation in the strains. The only thing that changed during the evolution experiment is the yeast extract (EMD millipore, Burlington, USA), for which the lot number changed. Further testing suggests that the pink/red coloration of the ade2-Δ mutants is media dependent. On one of the yeast extract lot used, white colonies appear red and show slower growth while on the other, most colonies are white and show normal growth. The slower growth is common to all ade2-Δ strains (Supplementary Figure 22)”

Reviewer #3 (Remarks to the Author):

3.1 Charron et al constructed interspecies hybrids between *Saccharomyces* species with various levels of genome sequence divergence. They then passaged the hybrids through ~800 mitotic generations on solid agar media, with random colony choice (bottlenecking) to minimize selection. They analyzed the fertility (meiotic fitness) of each hybrid at the beginning, middle and end of the experiment. Most of the hybrids had low fertility and there was no significant trend in how fertility evolved (equal numbers of lines decreased and increased their fertility; Fig. 2B). However the major result, emphasized in the manuscript's title, is the discovery that a small proportion of the hybrids regained high fertility by undergoing spontaneous duplication of their whole genome. That is, the initial cross was $1n \times 1n$ (haploids from two parental strains mated), forming a $2n$ hybrid (containing one copy of the genome from each parental species), which then spontaneously became $4n$ and fertile (with 2 copies of the genome from each parental species). The dramatic increase in fertility is attributed to the ability to pair homologous rather than homeologous (sequence-divergent) chromosomes. The proportion of hybrids showing this change was about 3% (7 of 214 hybrids examined).

Answer 3.1: We thank the reviewer for his valuable comments.

3.2 The manuscript's strengths are (1) that it provides a direct demonstration that meiotic (sexual) fertility can be restored during a period of mitotic (asexual) growth in yeast hybrids, and (2) that it shows that genome doubling is the most potent mechanism of fertility restoration. Moreover, (3) it shows that fertility can be regained very quickly in evolutionary terms (a few hundred mitotic generations). These results are consistent with current opinions about how the ancient whole-genome duplication (WGD) occurred during evolution of an ancestor of budding yeasts. Furthermore, the process that Charron et al observed in yeast should in principle be applicable to any unicellular eukaryotic species that, like yeast, can reproduce both vegetatively or sexually, so the process may be a widespread evolutionary mechanism.

Answer 3.2: We thank the reviewer for acknowledging the strengths of our work.

3.3 The manuscript's weaknesses are that that it provides little new insight into the *molecular mechanism* of fertility restoration. The observation that WGD restores fertility in the hybrids is not particularly surprising because Greig and colleagues (ref. 21) previously showed that artificially causing WGD in an interspecies hybrid restores its fertility. The surprise in Charron et al's work is that WGD occurs spontaneously at a significant frequency. But what is the mechanism of spontaneous WGD?

Answer 3.4: We thank the reviewer for raising concerns in order to improve our manuscript.

3.5 What this manuscript lacks is insight into the mechanism of spontaneous WGD. Such insight could be obtained by more analysis of the genomes of the fertile interspecies hybrids made by Charron et al. Previous analyses of the genomes of fertile natural interspecies *Zygosaccharomyces* hybrids proposed that their mechanism of fertility restoration was damage to one copy of the MAT locus (PMIDs 28510588 30052970 28842546). This process essentially converts a hybrid zygote into a gamete, enabling it to mate. Did something similar happen in the fertility restoration events that Charron et al observed? My opinion is that the current manuscript is incomplete without an examination of the MAT loci of the fertility-restored strains.

Answer 3.5: We thank the reviewer for this comment. To investigate the mechanism of whole genome doubling observed in the tetraploid hybrids we sequenced the whole genome of all tetraploids at Tini and Tend. By combining read coverage and allele frequency analysis, we tested the different possibilities of mechanisms that could cause damage to the MAT locus. We observed some cases of aneuploidies that affect only one parental copy of the MAT locus and LOH that affect the silent HML locus. We added the main results of these analysis in the text (lines 239-270) :

“As stated above, whole genome doubling could occur either by endoreduplication which is a consequence of cytokinesis failure³⁴ or by means of damage to one copy of the MAT locus in the hybrid^{30,31} (Supplementary Figure 15). This damage to the MAT locus could cause hybrid cells to behave as a haploid, switch mating type and hence autotetraploidize. In this experiment, mating type switching may not occur using the standard process because the necessary HO gene was deleted. The main way by which autotetraploidization could occur by mating in our study is to have two hybrids with damage to the opposite MAT loci that are in the same colony and are close enough to mate with each other (Supplementary Figure 15), which is a very unlikely event. To investigate this, we sequenced the genome of the 8 tetraploid lines (5 L_{div} , 2 M_{div} and 1 H_{div}) at T_{ini} and T_{end} . We indeed found that the frequency of parental alleles across the genomes are roughly 50%, showing that the strains are not aneuploids that would have DNA content equivalent to tetraploidy (Fig. 3D, Supplementary Figure 16 and 17). One exception is observed for the M1_40 line that show allele frequency corresponding to a triploid state at T_{end} (Fig. 3D, Supplementary Figure 16) while GBS data show a tetraploid state at T_{mid} and a diploid state at T_{end} (Supplementary Figure 14). A different colony was isolated each time from these time-points. This is again consistent with segregating ploidy and colony heterogeneity which are probably due to extreme genomic instability in this particular line (for more details see supplementary text). Next, we investigated the total or partial chromosome loss or LOH of the MAT locus region (Supplementary Figure 15). We identified only two tetraploids with aneuploidy on chromosome III containing the MAT locus (Supplementary Figures 18, 19 and 20). However, these aneuploidies affect only one copy of the mating type (Supplementary text and Supplementary Figure 21). We thus find no evidence of

damage to the *MAT* locus of tetraploid hybrids that could have caused mating between diploid hybrids. Thus, these results suggest that endoreduplication is the most likely mechanism of whole genome doubling. However, we cannot exclude that damage to the *MAT* loci occurring by the loss of chromosome III could be undetectable because this chromosome could have been regained following tetraploid formation. Chromosome III was indeed found to be the most unstable chromosome among *S. cerevisiae* diploids, triploids and *S. cerevisiae*-*S. bayanus* hybrids³⁵. Furthermore, transient dynamics of aneuploidy were repeatedly observed in yeast laboratory evolution experiments under stressful conditions^{37,38}. Such successive aneuploidies could happen in our hybrids lines because aneuploidy is prevalent (Fig. 3C, Supplementary Figure 18) and tetraploids are notorious for showing genome instability³⁶.”

-More details and figures (Fig. S15, S16, S17, S18, S19, S20 and S21) in the supplementary data (lines 28-62).

“Supplementary text

Mechanisms of whole genome duplication

To investigate the potential mechanisms of whole genome doubling that led to tetraploid hybrids, by endoreduplication or mating between two diploid hybrids, we sequenced the genome of all tetraploids at T_{ini} and T_{end} . The main way by which autotetraploidization could occur by mating in our study is to have two hybrids with damage to the opposite *MAT* loci in proximity to mate. Such tetraploid hybrids should thus have two *MAT* loci of opposite mating type with damaged function. By combining read coverage and allele frequency analysis, we investigated the presence of total or partial loss of chromosome III containing the *MAT* locus (Supplementary Figures 18 and 19) or LOH events around the *MAT* locus (Supplementary Figure 20). We identified two tetraploids that show aneuploidy on chromosome III at T_{end} . One of the L1 tetraploids has one more copy of chromosome III from the SpC parent and the H2 tetraploid has lost one copy from the *S. cerevisiae* parent. In both cases, aneuploidy affects only one copy of the mating type. Considering the fact that total or partial chromosome III loss should affect the two mating type loci in the tetraploid hybrid, these results show that the loss of chromosome III is not the molecular mechanism leading to whole genome doubling. Allele frequency analysis confirms these aneuploidies and show that there is no LOH events around the *MAT* locus in all tetraploids.

We also examined the copy number of *MAT α* and *MAT β* sequences from each parent (figure S21). For an accurate analysis, we had to take in account the silent mating type copies. Indeed, in addition to the *MAT* locus, *Saccharomyces* yeasts carry two unexpressed, but complete, copies of mating-type genes at the silent loci, *HML* and *HMR* localised also on chromosome III (Supplementary Figure

20)¹. The HML locus carries MAT α sequence while HMR carries MATa sequence. Thus, knowing that the SpB parent is MATa and the SpC, SpA and S. cerevisiae parents are MAT α , each hybrid carries both mating type copies from each parent with a different copy number of MATa or MAT α corresponding to each parental origin.

Copy number and allele frequency variations observed in MATa/MAT α copies from each parent (Supplementary Figure 21) are due either to the aneuploidies mentioned above (L1_51) (Supplementary Figure 18) or to an LOH in the HML locus (L2_36) (Supplementary Figure 20) or a combined effect of aneuploidy and LOH in the HML locus (H2_38) (Supplementary Figures 18, 20 and 21) which could not have led to mating.

Our results show no evidence of damages in the MAT locus in tetraploid hybrids that could have caused mating between diploid hybrids. The cases of aneuploidies or LOH observed affect only one parental copy of the MAT locus or the silent HML locus. Also, there is no evidence for damaging mutations occurring in the MAT loci. Thus, according to these results, autodiploidization is the most likely mechanism of whole genome doubling.”

Supplementary Figure 15. Major potential mechanisms of whole-genome doubling

Whole genome doubling could occur by (1) autodiploidization, which is a consequence of cytokinesis failure. It could also be caused by means of damage to one copy of the MAT locus. The MAT locus damage could be caused by (2) partial or complete chromosome loss of the chromosome containing the MAT locus (chromosome III), (3) loss of heterozygosity around the MAT locus, (4) or an inactivating mutation in one copy of the MAT locus.

Supplementary Figure 16. The tetraploid hybrids result from a whole genome duplication of both parental genomes. Allele frequencies along the 16 chromosomes of the tetraploid hybrids at T_{end} are around 50%, similar to what is seen for the same diploid hybrids at T_{ini} . The heatmap represents allele frequencies after mapping reads generated by whole genome

sequencing on the *S. paradoxus* reference genome (MSH604) (276003 markers for L1 lines, 280756 markers for L2 lines, 471547 markers for M1 lines and 981580 markers for the H2_38).

Supplementary Figure 17. The *S. paradoxus* × *S. cerevisiae* (H2) hybrids segregate a copy of each parental genome in its diploid spores.

Spores from the tetraploid line H2_38 were plated on selective media to assess the segregation, during meiosis, of the selection cassettes that were introduced at the HO locus. The *S. cerevisiae* genome harbor the NAT resistance while *S. paradoxus* genome harbor the G418 resistance at the HO locus (Chr IV). Empty spaces on plates are spores that didn't form colonies on the original dissection plate. As all the viable spores inherited both resistances, it is likely that the spores contain full non-recombined *S. cerevisiae* and a *S. paradoxus* haplotypes.

Supplementary Figure 18. The loss of chromosome III is not the molecular mechanism leading to whole genome doubling. Sequencing read depth for bins of 10kb on the 16 chromosomes of the 8 tetraploid lines at T_{ini} (the left panel) and T_{end} (the right panel). The red line represents the average sequencing read depth of the whole genome. Several aneuploidies are detected in almost all hybrids at T_{ini} and T_{end} .

Supplementary Figure 19. The partial loss of Chromosome III is unlikely to be the molecular mechanism leading to whole genome doubling. Sequencing read depth for each bin of 100bp on chromosome III for the 8 tetraploid lines at T_{ini} (the left panel) and T_{end} (the right panel). The red lines represent the average sequencing read depth of the whole genome. The active mating type locus (MAT) is indicated by dashed lines.

Supplementary Figure 20. Loss of heterozygosity around the MAT locus is most likely not the molecular mechanism causing whole genome doubling. Allele frequencies along the chromosome III for the 8 tetraploid lines at T_{ini} (left panel) and T_{end} (right panel). The active mating type locus (MAT) and the silent mating type loci (HML and HMR) are indicated by dashed lines. The heatmaps represent allele frequency after mapping on *S. paradoxus* reference genome (6611 markers for L1 lines, 5986 markers for L2 lines, 11794 markers for M1 lines and 20537 markers for the H2_38).

Supplementary Figure 21. Copy number variation and allele frequency of MATa and MAT α sequences of tetraploid hybrids show no double damage in the MAT locus.

(A) The bar plots represent the ratio of MATa and MAT α average read depth for the 8 tetraploid lines at T_{ini} and T_{end} as well as the 6 haploid parental strains. The average sequencing read depth of MATa and MAT α were calculated for a sequence of 1.3 kb containing SNPs that differentiates the two copies. (B) The heatmaps show the average allele frequencies (AF) of the MATa and MAT α sequences for the 8 tetraploid lines at T_{ini} and T_{end} . After mapping reads on *S. paradoxus* MATa and MAT α reference sequences, the average AF were calculated for a sequence of 1.3 kb containing SNPs that differentiate the two mating type copies. The panel on the left shows the average allele frequencies of both mating type sequences corresponding to the haploid SpB parent (which is MATa) alleles and the panel on the right those corresponding to the haploid SpC (for L1 and L2 lines), SpA (for M1 lines) or *S. cerevisiae* (for H2_38) parents (which are MAT α) alleles.

Answer 3.6: We added a comment on this in the text (lines 340-343):

“The rate of fertility restoration by genome doubling observed in our experiments might be an underestimate of the rate that would occur in yeast hybrids that contain the HO gene. In the event of damage to one of the mating type loci, the wild type HO would allow the mating type switch and self mating in hybrids, leading to a duplicated genome.”

Minor comments

3.7 L106, This point was recently confirmed by Rogers et al, PMID 30419022.

Answer 3.7: We changed the sentence to better represent current knowledge (lines 178-180).

*“As infertility of *Saccharomyces* hybrids is mainly due to anti-recombination caused by the mismatch repair machinery acting on homeologous chromosomes pairs (Greig, Travisano et al. 2003, Rogers 2018), [...]”*

3.8 L109-113. The text here is not very clearly written and I think that readers would struggle to understand the difference between the two explanations, particularly because line 112 says they are “equivalent”. Is explanation #1 an endo-reduplication model, i.e. DNA replication without cell division? In explanation #2, are you proposing that the spores have 2n ploidy, i.e. that they were produced without meiosis? Perhaps a picture would help.

Answer 3.8: We agree that this part of the text could have been better explained. We therefore changed this part. The first explanation was indeed endo-reduplication. But the second is sporulation followed by intratetrad mating between two spores (which would be equivalent to the autodiploidization we performed if the two spores were very similar genetically, which is unlikely). We also added a third hypothesis which is the one suggested in comment 3.5 (lines 184-205).

*“The first potential mechanism would be an endoreduplication event, i.e. spontaneous chromosome doubling following a failed cell division during mitosis³⁰. Such an event would lead to the production of identical homologues that would restore correct chromosome segregation. The second mechanism would be damage to a copy of the MAT locus that would convert the diploid hybrid into behaving as a gamete. Two such diploid gametes could then mate, generating a fertile tetraploid hybrid. This path to fertility recovery was recently observed in hybrid species of the *Zygosaccharomyces* genus^{31,32}, so it is in principle an accessible path to fertility recovery. However, this would need rare events to co-occur in the same colony and to produce two diploid gametes of opposite mating type. The last potential mechanism would be that strains could have sporulated during the experiment and spores of opposite mating types could have mated. This would be the equivalent of our ITC lines where sometimes a single cross between*

two spores can bring fertility back to high values (Supplementary Figure 6). This third option is very unlikely because sporulation happens under very specific environmental conditions, principally nitrogen starvation. The frequency of streaking to fresh media during the experiment would prevent such depletion to happen. In addition, this scenario would often lead to spores that are aneuploid, making fertility recovery unlikely even after mating. All these mechanisms would generate strains with increased spore viability, but the lines are expected to show a change from diploidy to tetraploidy in the first two scenarios, allowing to differentiate these mechanisms. Mitotic loss of heterozygosity could also be involved but would not be expected to lead to such dramatic recovery of fertility³³. We tested these potential mechanisms by measuring the total cellular DNA content of the lines to infer ploidy, genome-wide genotyping and whole-genome sequencing of some of the strains.”

3.9 Line 114 says that you “tested these hypotheses”, but the test you carried out is a test of whether the (identical) prediction of both of the hypotheses (i.e. 4n ploidy) is verified. The hypotheses were not tested individually. In fact, the MAT locus analysis I suggested above could differentiate between them.

Answer 3.9: We now tested these hypotheses by analyzing the MAT locus as proposed. We thus sequenced the whole genome of all tetraploids at Tini and Tend. We added the main results of these analysis in the text and more details and figures in the supplementary data (Figure S15, S16, S18, S19, S20 and S21) (see above, comment 3.5).

3.10 L102 mentions 7 mitotic lines with high fertility (4L, 2M, 1H), but L136 then refers to “6 lines showing more than 70% fertility after evolution”. Looking at Fig 4, it’s not immediately obvious which lines are the 6 in question. It seems that one of the M lines had high fertility and was 4n at the middle timepoint, but at the end timepoint it lost fertility and its ploidy changed (??). If this is correct, the ploidy decrease in this strain needs some more explanation/discussion. It would help if you named the 4n strains.

Answer 3.10: We corrected this sentence to include the eight strains that displayed spectacular or significant fertility recovery (lines 230-232):

“...the 7 lines that displayed significant spectacular fertility recovery and one L line that showed significant fertility improvement, but of lower magnitude, during evolution became tetraploid”

-One of the M lines indeed was 4n with high fertility at Tmid and then at Tend lost fertility and changed ploidy to 2n. Two possibilities could explain this result. The first one is that the isolated colony at Tmid might be heterogeneous containing diploid and tetraploid cells. In this

case, during the preculture for glycerol stock the tetraploid cells might out-compete diploid ones explaining why we observe only tetraploids at T_{mid} . However, the colony isolated during the subsequent passage was a diploid one explaining why we observe only diploids in subsequent passages. The other possibility is that the tetraploid strain at T_{mid} corresponds to another M1 line than M1_40. However, we did not detect any evidence to confirm that. We tried to check that using whole genome sequencing and GBS data. The presence of an aneuploidy in Chromosome I of the M1_40 line at T_{ini} and T_{end} indicates that it is the same strain. Also, the two tetraploids M1_40 and M1_49 are different because the M1_49 has also specific aneuploidies which discredits the possibility of having contaminated the M1_40 by M1_49. We added a short discussion about this in the text (lines 233-236):

“One of the two tetraploid M1 lines (M1_40) returned to diploidy at T_{end} but this is more likely to be due to segregating ploidy and colony heterogeneity at T_{mid} rather than return to diploidy (for more details see Supplementary text).”

We also added a supplementary data's section “Mechanisms of whole genome duplication” (lines 64-80).

“Return to diploidy after tetraploidy

These analyses allowed us to examine the genome of the M1 tetraploid (M1_40) whose ploidy and fertility increased at T_{mid} and then decreased at T_{end} . This result could be explained by the presence of heterogenous colonies during our evolution experiment. The isolated colony at T_{mid} might segregating diploid and tetraploid cells. One type may have fixed in the glycerol stock and be lost in the next round of propagation, explaining why we observe only tetraploids at T_{mid} . However, the colony isolated during the subsequent passage was a diploid one explaining why we observe only diploids in subsequent passages.

Besides, GBS data are consistent with ploidy and fertility data for M1_40 line, however genome sequencing show a triploid state at T_{end} instead of diploid. The detection of chromosome I loss of the SpA parent copy at T_{ini} and T_{end} (Supplementary Figure 18) confirms that both sequenced isolated colonies correspond to the same M1_40 line at two different times. The same variations are observed for the M2_36 line at T_{ini} that show a different ploidy state between Ploidy and GBS data ($2n$) and sequencing data ($4n$, with many aneuploidies) while the same LOH is observed in chromosome VIII at T_{ini} and T_{end} (Supplementary Figure 16). These results suggest that there is still heterogeneity among isolated colonies from the glycerol stock probably due to genomic instability of hybrids.”

3.11 L148: “Mitotic fertility recovery contrasts with ... results ... showing that sexual reproduction leads to ... recovery of fertility”. Why do you call this a contrast? There is no reason to expect the two processes to be mutually exclusive.

Answer 3.11: The contrast resided in the timeframe of the recovery. The recovery of fertility by meiotic recombination seems like a more gradual process (Fig. S6A) while the mitotic recovery by genome duplication is punctuated. Also, the contribution of mitotic recombination in the recovery of fertility, which is very minor, can also be contrasted to the contribution of meiotic recombination. We tried to clarify this part by changing the sentence (lines 317-319):

“This would suggest that the contribution of mitotic recombination to the recovery of fertility in hybrid lineages could be a slow process compared to meiotic recombination and whole-genome duplication.”

3.12 In Fig 3C, why are only four 4n strains are shown, of the 6-7 that were identified? How are we supposed to know which ones are shown?

Answer 3.12: In this figure we show only allele frequency of the *SpC* parent in the L cross to show that the triploids have two copies of the *SpC* genome and diploids and tetraploids have one or two copies respectively of each parental genome. The allele frequency of the other tetraploids are shown in the supplementary figure S13 and figure S14 because they have different parental strains (*SpBx SpA* and *SpBxScer*). We specified this in the figure legend:

“(C) Frequency of 171 markers corresponding to the SpC parent alleles across the genome of a subset (6 diploids, 6 triploids and all the 5 tetraploids) of L_{div} lines show around 50% of SpC alleles in the diploid and tetraploid strains and 66% in triploid strains.”

We also added two figures (Figure 3D and Figure S16) to show the allele frequencies estimated by whole-genome sequencing across the 16 chromosomes of the eight tetraploids.

Fig. 3. (D) Allele frequencies estimated by whole-genome sequencing across the 16 chromosomes. Because diploid hybrids have one copy of each genome, allele frequencies are centered around 50% at T_{ini} . This frequency is preserved in most tetraploids, showing the all chromosomes were doubled at T_{end} .

“Supplementary Figure 16. The tetraploid hybrids result from a whole genome duplication of both parental genomes.

Allele frequencies along the 16 chromosomes of the tetraploid hybrids at T_{end} are around 50%, similar to what is seen for the same diploid hybrids at T_{ini} . The heatmap represents allele frequencies after mapping reads generated by whole genome sequencing on the *S. paradoxus* reference genome (MSH604) (276003 markers for L1 lines, 280756 markers for L2 lines, 471547 markers for M1 lines and 981580 markers for the H2_38).”

Reviewers' Comments:

Reviewer #1:

Remarks to the Author:

The authors have thoroughly revised the manuscript and addressed each of my comments in their response letter. I can recommend publication of this article.

Rike Stelkens

Reviewer #2:

Remarks to the Author:

I thank the authors for their clear rebuttal letter, it is among the easiest to parse that I have seen. I am satisfied with the majority of revisions they have made in response to my initial comments.

I have only a couple of follow-up comments that pertain to the revision.

Answer 2.4 - Strain extinction and intrapopulation variability

I'm surprised that you weren't able to recapitulate strain extinction from the new experiment that was done (I suspect the authors were as well). I appreciate this experiment and think it was a good test. I would think this result is worth mentioning somewhere in the manuscript even if it is not the result that was expected. It may be (as posited) that the glycerol step is adding a major point of selection, and this alone to me is worth documenting somewhere other than in a rebuttal letter. I personally have wondered about the influence of "pre-culture"/culture-from-frozen" step on fitness assays etc. Of minor related note, in Figure 1E the Tmid results are not shown.

Answer 2.6 - Diploidization of ancestral culture

In a similar vein, I found the answer (and additional experimental results) interesting and worth discussing outside of a rebuttal letter.

Reviewer #3:

Remarks to the Author:

The authors have carried out an extensive analysis of the MAT locus status in their strains, and have comprehensively shown that MAT locus damage was not involved in the genome duplications they observed. I am satisfied with the new data they present and the changes they made to the manuscript, and have no further comments.

REVIEWERS' COMMENTS:

Reviewer #1 (Remarks to the Author):

The authors have thoroughly revised the manuscript and addressed each of my comments in their response letter. I can recommend publication of this article.
Rike Stelkens

Answer: We thank Dr Stelkens for her overall positive evaluation of our work and recommendation for publication.

Reviewer #2 (Remarks to the Author):

I thank the authors for their clear rebuttal letter, it is among the easiest to parse that I have seen. I am satisfied with the majority of revisions they have made in response to my initial comments.

Answer: We thank the reviewer for the favorable review.

I have only a couple of follow-up comments that pertain to the revision.

Answer 2.4 - Strain extinction and intrapopulation variability

I'm surprised that you weren't able to recapitulate strain extinction from the new experiment that was done (I suspect the authors were as well). I appreciate this experiment and think it was a good test. I would think this result is worth mentioning somewhere in the manuscript even if it is not the result that was expected. It may be (as posited) that the glycerol step is adding a major point of selection, and this alone to me is worth documenting somewhere other than in a rebuttal letter. I personally have wondered about the influence of "pre-culture"/culture-from-frozen" step on fitness assays etc. Of minor related note, in Figure 1E the Tmid results are not shown.

Answer: We added mention to this result (lines 108-112):

“However, replicating multiple colonies from the last available glycerol stocks of some extinct L_{div} and H_{div} lineages over 4 passages (80 generations) could not recapitulate the high extinction rate we observed during the experimental evolution. This may indicate that the pre-cultures preceding the glycerol stock constitute a selection step favoring the most genetically stable individuals within the colony, even if they are in minority.”

Answer 2.6 - Diploidization of ancestral culture

In a similar vein, I found the answer (and additional experimental results) interesting and worth discussing outside of a rebuttal letter.

Answer: We added a discussion about diploidization of SpC parents in the main text (lines 234-243):

“The proportion of triploids among hybrids may depend on the initial proportion of diploid parents (pseudohaploid) in the preculture used for the crosses, which itself is stochastic (Supplementary Figure 9). The frequency of SpC diploid parents may also depend on whether it appears early or late in the cell culture. The absence of triploids in some other L_{div} crosses indicate that diploidization in parental strains is background dependent. Phenotypic, genomic and karyotypic diversity have been observed among strains of the SpB and SpC lineages^{17,19}. These results suggest that some SpC haploid strains may be prone to spontaneous genome doubling, the origin of which will need more investigations. For instance, variation that affect key genes involved in cell cycle regulatory pathways could be on the origin of this process.”

Reviewer #3 (Remarks to the Author):

The authors have carried out an extensive analysis of the MAT locus status in their strains, and have comprehensively shown that MAT locus damage was not involved in the genome duplications they observed. I am satisfied with the new data they present and the changes they made to the manuscript, and have no further comments.

Answer: We thank the reviewer for the favorable review.